# Perturbation biology nominates upstream–downstream drug combinations in RAF inhibitor resistant melanoma cells

**Anil Korkut[1]\*, Weiqing Wang[1]\*, Emek Demir[1], Bülent Arman Aksoy[1,2], Xiaohong Jing[1], Evan J Molinelli[1], Özgün Babur[1], Debra L Bemis[1], Selcuk Onur Sumer[1], David B Solit[3,4], Christine A Pratilas[5], Chris Sander[1]\***

[1]Computational Biology Center, Memorial Sloan Kettering Cancer Center, New York, United States; [2]Tri-Institutional Training Program in Computational Biology and Medicine, New York, United States; [3]Human Oncology and Pathogenesis Program, Memorial Sloan Kettering Cancer Center, New York, United States; [4]Department of Medicine, Memorial Sloan Kettering Cancer Center, New York, United States; [5]The Sidney Kimmel Comprehensive Cancer Center at Johns Hopkins University, Baltimore, United States

**Abstract** Resistance to targeted cancer therapies is an important clinical problem. The discovery of anti-resistance drug combinations is challenging as resistance can arise by diverse escape mechanisms. To address this challenge, we improved and applied the experimental-computational perturbation biology method. Using statistical inference, we build network models from high-throughput measurements of molecular and phenotypic responses to combinatorial targeted perturbations. The models are computationally executed to predict the effects of thousands of untested perturbations. In RAF-inhibitor resistant melanoma cells, we measured 143 proteomic/phenotypic entities under 89 perturbation conditions and predicted c-Myc as an effective therapeutic co-target with BRAF or MEK. Experiments using the BET bromodomain inhibitor JQ1 affecting the level of c-Myc protein and protein kinase inhibitors targeting the ERK pathway confirmed the prediction. In conclusion, we propose an anti-cancer strategy of co-targeting a specific upstream alteration and a general downstream point of vulnerability to prevent or overcome resistance to targeted drugs.

**For correspondence:** bpmel@sanderlab.org (AK, WW, CS)

**Competing interests:** The authors declare that no competing interests exist.

## Introduction

### Drug resistance in cancer treatment

The inhibition of key oncogenes with target-specific agents elicits dramatic initial response in some cancers such as prostate cancer and melanoma (*Bollag et al., 2010*; *Clegg et al., 2012*). Even for the most successful single-agent targeted therapies, however, drug resistance eventually emerges leading to rapid progression of metastatic disease (*Garraway and Janne, 2012*). The mechanism of drug resistance may involve selection of resistant sub-clones, emergence of additional genomic alterations, and compensating interactions between alternative signaling pathways (*Choi et al., 2007*; *Huang et al., 2011*; *Wagle et al., 2013*). One potential solution to overcome drug resistance is to combine targeted drugs to block potential escape routes (*Fitzgerald et al., 2006*). Therefore, there is currently a need for systematic strategies to develop effective drug combinations.

### Drug resistance in melanoma

Targeted therapy has been particularly successful in treatment of melanoma. *BRAFV600E* gain-of-function mutation is observed in ~50% of melanomas (*Davies et al., 2002*). Direct inhibition of BRAFV600E by the RAF inhibitor (RAFi) vemurafenib and inhibition of the downstream MEK and ERK

**eLife digest** Drugs that target the activity of specific genes could potentially form precise cancer therapies. Some cancers, including the aggressive skin cancer called melanoma, initially respond well to such treatments. However, resistance to drugs develops quickly, leading to the rapid regrowth of the tumors.

Resistance can develop in a number of ways. For example, to prevent the drug from working or to compensate for the effects of a drug, cancer cells can adapt their signaling processes or acquire genetic mutations or other modifications that affect how genes are expressed.

A well-designed combination of drugs that targets multiple molecular pathways can make it harder for cells to resist treatment, as this limits the number of available 'escape' pathways that bypass the drug targets. However, it is difficult to accurately predict how a cell will respond when treated with a particular drug, making it extremely challenging to design effective drug combinations.

In 2013, researchers developed a technique to build predictive models of cellular response pathways based on data collected from perturbation experiments followed by mathematical modeling. Now, Korkut et al.—including several of the researchers involved in the 2013 work—have refined this technology and applied it to the problem of preventing drug resistance in cancer cells.

Computer simulations that used the mathematical models suggested a particular strategy of 'upstream–downstream targeting' in cells that were insensitive to the clinically successful drug vemurafenib (an inhibitor of RAF proteins, which are often mutated in cancers). In the landscape of signaling pathways, the target of the upstream drug is on or near the mutated RAF protein. c-Myc, the indirect target of the downstream drug helps to express genes that trigger signals that cause the cells to grow. Inhibiting both targets in parallel may have the dual advantage of blocking the activation of the tumor-specific growth pathway while reducing the cancer cells' attempts to bypass the activation block.

An initial test of the designed drug combination required moving from computer simulations to the laboratory using cell cultures originally derived from melanoma tumors. When Korkut et al. applied the drug combination, the combined treatment successfully blocked cell growth. The results suggest that the data-driven computer modeling strategy termed perturbation biology could be a useful tool for identifying effective cancer drug combinations for further preclinical research, possibly followed by clinical trials.

kinases have yielded response rates of more than 50% in melanoma patients with this mutation (*Chapman et al., 2011*; *Flaherty et al., 2012b*). At the cellular level, inhibition of the ERK pathway leads to changes in expression of a set of critical cell cycle genes (e.g., *CCND1, MYC, FOS*) and feedback inhibitors of ERK signaling (e.g., *DUSP, SPRY2*) (*Pratilas et al., 2009*). Resistance to vemurafenib emerges after a period of ~7 months in tumors that initially responded to single-agent therapy (*Chapman et al., 2011*; *Sosman et al., 2012*). Multiple RAFi and MEKi (e.g., PD-0325901, Trametinib) resistance mechanisms, which may involve alterations in NRAS/ERK pathway (e.g., NRAS mutations, switching between RAF isoforms) or parallel pathways (e.g., PTEN loss), have been discovered in melanoma (*Johannessen et al., 2010*; *Nazarian et al., 2010*; *Poulikakos et al., 2010*; *Xing et al., 2012*).

The alterations associated with drug resistance may pre-exist alone, in combinations, or emerge sequentially and vary substantially between patients (*Van Allen et al., 2014*). Effective drug combinations may target diverse resistance mechanisms. Despite anecdotal success, conventional methods and combinatorial drug screens generally fail to come up with effective combinations due to the genomic complexity and heterogeneity of tumors (*Zhao et al., 2014*). In order to more effectively nominate drug combinations, we propose to employ system-wide models that cover interactions between tens to hundreds of signaling entities and can describe and predict cellular response to multiple interventions. There have been prior attempts to construct such signaling models. De novo and data-driven quantitative models were able to cover only a few signaling interactions and therefore had limited predictive power (*Nelander et al., 2008*; *Bender et al., 2011*; *Klinger et al., 2013*; *Oates et al., 2014*). Qualitative or discrete models can cover more interactions but typically lack the capability of generating quantitative predictions (*Saez-Rodriguez et al., 2009*; *Breitkreutz et al., 2010*; *Saez-Rodriguez et al., 2011*). Detailed physicochemical models derived using generic biochemical kinetics data can be quite comprehensive and quantitative but typically lack sufficient cell-type specificity required for translationally useful predictions (*Chen et al., 2009*).

We construct comprehensive, cell-type specific signaling models that quantitatively link drug perturbations, (phospho)proteomic changes, and phenotypic outcomes (*Figure 1*). The models capture diverse signaling events and predict cellular response to previously untested combinatorial interventions. In order to generate the training data for network modeling, we first perform systematic perturbation experiments in cancer cells with targeted agents. Next, we profile proteomic and phenotypic response of cells to the perturbations. The cell-type specific response data serve as the input for network inference. In this study, we also incorporate prior pathway information from signaling databases to narrow the parameter search space and improve the accuracy of the models. For this purpose, we developed a computational tool (Pathway Extraction and Reduction Algorithm [PERA]) that automatically extracts priors from the Pathway Commons signaling information resource (*Demir et al., 2010*; *Cerami et al., 2011*).

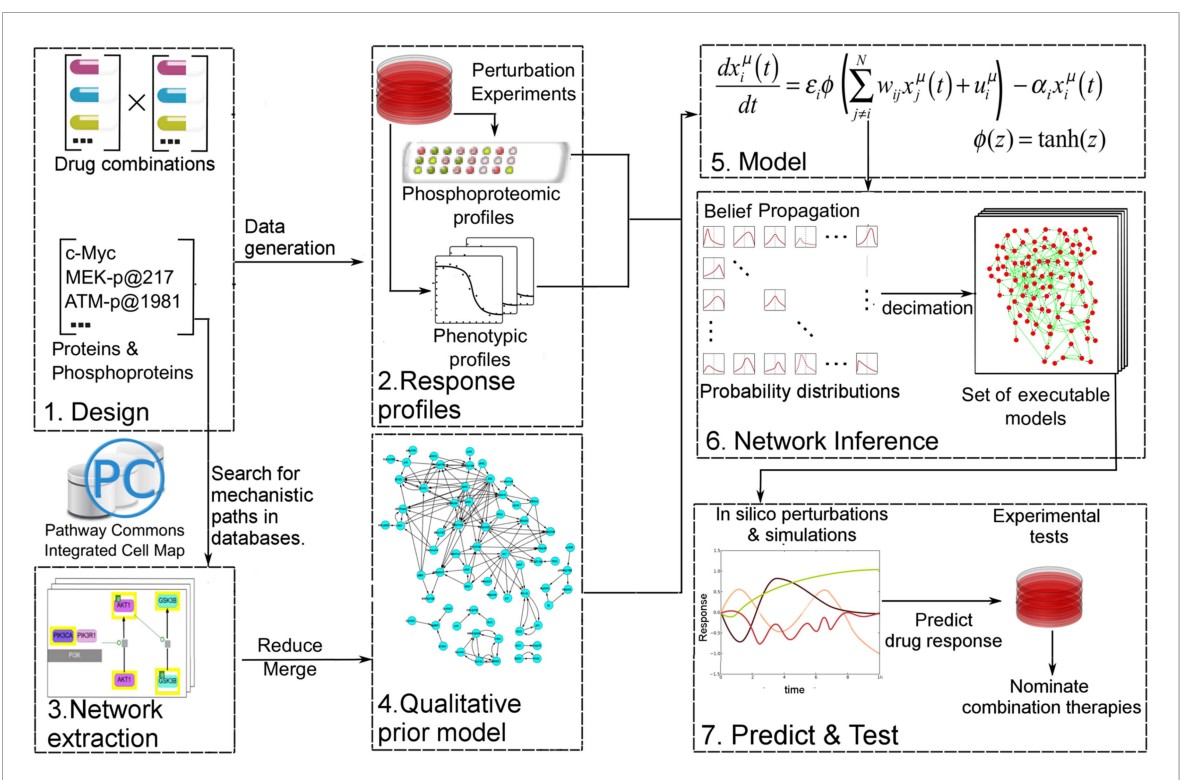

**Figure 1**. Quantitative and predictive signaling models are generated from experimental response profiles to perturbations. Perturbation biology involves systematic perturbations of cells with combinations of targeted compounds (Box 1, 2), high-throughput measurements of responses (Box 2), automated extraction of prior signaling information from databases (Box 3, 4), construction of ordinary differential equation (ODE)-based signaling pathway models (Box 5 and *Equation 1*) with the belief propagation (BP) based network inference algorithm (Box 6), and prediction of system responses to novel perturbations with the models and simulations (Box 7) (*Molinelli, Korkut, Wang et al., 2013*). The newly developed 'Pathway Extraction and Reduction Algorithm' (PERA) (Box 3) generates a qualitative prior model (Box 4), which is a network of known interactions between the proteins of interest (i.e., profiled (phospho)proteins). This is achieved through a search in the Pathway Commons information resource, which integrates biological pathway information from multiple public databases. In the quantitative network models, the nodes represent measured levels of (phospho)proteins or cellular phenotypes, and the edges represent the influence of the upstream nodes on the time derivative of their downstream effectors. This definition corresponds to a simple yet efficient ODE-based mathematical description of models (Box 5). Our BP-based modeling approach combines information from the perturbation data (phosphoproteomic and phenotypic) with prior information to generate network models of signaling (Box 6). We execute the resulting ODE-based models to predict system response to untested perturbation conditions (Box 7).

The following figure supplement is available for figure 1:

**Figure supplement 1**. BP-guided decimation algorithm.

Even in the presence of large training data and priors, network inference is a difficult problem due to the combinatorial complexity (i.e., exponential expansion of the parameter search space with linear increase of parameters). For example, to infer a network model with 100 nodes using Monte Carlo-based methods, we in principle would need to cover a search space that includes $\sim 2^{(100 \times 100)}$ network models—a computationally impossible task. To circumvent this problem, we previously had developed a network modeling algorithm based on belief propagation (BP), which replaces exhaustive one-by-one searches over many individual network models by a search over probability distributions representing sets of low-error network models (*Molinelli, Korkut, Wang et al., 2013*). The algorithm enables us to construct models that can predict response of hundreds of signaling entities to any perturbations in the space of modeled components. Here, we improved the perturbation biology method through automated incorporation of prior information (signaling interactions from databases) to obtain more accurate network models. The prior information is extracted from the signaling databases using the newly developed PERA tool. To improve predictive power and preserve cell-type specificity, we use prior information as soft restraints on search space through use of a probabilistic error model of priors, that is, the algorithm rejects interactions that do not conform to the experimental training data and predicts novel interactions not sampled in the priors (see *Equations 2–7*). To derive richer and more informative network models, we also scaled up the implementation to deal with a larger number of proteomic and phenotypic nodes (see 'Materials and methods' for details and comments on potential future opportunities in perturbation biology). To quantitatively predict cellular response to combinatorial perturbations, we simulate the fully parameterized network models with in silico perturbations until the system reaches steady state (*Figure 1*). The steady-state readout for each proteomic and phenotypic entity (i.e., system variables) is the predicted response to the perturbations.

In this study, we improved and applied the perturbation biology method to devise a potentially generalizable strategy for overcoming resistance to targeted cancer therapies. We constructed cell-type specific network models of signaling from perturbation experiments in RAFi-resistant melanoma cells (SkMel-133 cell line). The melanoma cells used for network modeling have a *BRAFV600E* mutation and homozygous deletions in *PTEN* and *CDKN2A*. The models quantitatively link 82 (phospho)proteomic nodes (i.e., molecular concentrations) and 12 protein activity nodes with 5 cellular phenotype nodes (e.g., cell viability). As shown by cross-validation calculations, use of prior information significantly improved the predictive power of the models. Once the predictive power was established, we expanded the extent of the drug response information from a few thousand experimental data points to millions of predicted node values. Based on the predictions, which cannot be trivially deduced from experimental data, we nominated co-targeting of c-Myc and BRAF or MEK as a potential strategy to overcome RAFi-resistance. To test the prediction, we first experimentally showed that the BET bromodomain inhibitor, JQ1, reduces c-Myc expression. Next, we showed that JQ1 blocks the growth of RAFi-resistant SkMel-133 cells in synergy with RAF/MEK signaling inhibition, and in this context, overcomes the drug resistance. Based on these results, we put forward the falsifiable hypothesis that co-targeting a specific upstream alteration and a general downstream point of vulnerability is a good strategy to prevent or overcome resistance to targeted drugs.

## Results

### Experimental response maps from proteomic/phenotypic profiling

#### Drug perturbation experiments in melanoma cells

We performed systematic perturbation experiments in malignant melanoma cells (*Figure 2A*) to generate a rich training set for network inference. The RAFi-resistant melanoma cell line SkMel-133 was treated with combinations of 12 targeted drugs (*Figure 2A*, see *Supplementary file 1A* for drug targets and doses, *Supplementary file 1C* for all perturbation conditions). The perturbations consisted of systematic paired combinations and multiple doses of single agents. This procedure generated 89 unique perturbation conditions, which targeted specific pathways including those important for melanoma tumorigenesis such as ERK and PI3K/AKT (*Haluska et al., 2006*).

#### Proteomic/phenotypic profiles

A key aspect of the data acquisition for network inference is combining the proteomic and cellular phenotypic data so that the resulting models quantitatively link the proteomic changes to global cellular responses. Toward this objective, we profiled the melanoma cells for their proteomic and phenotypic

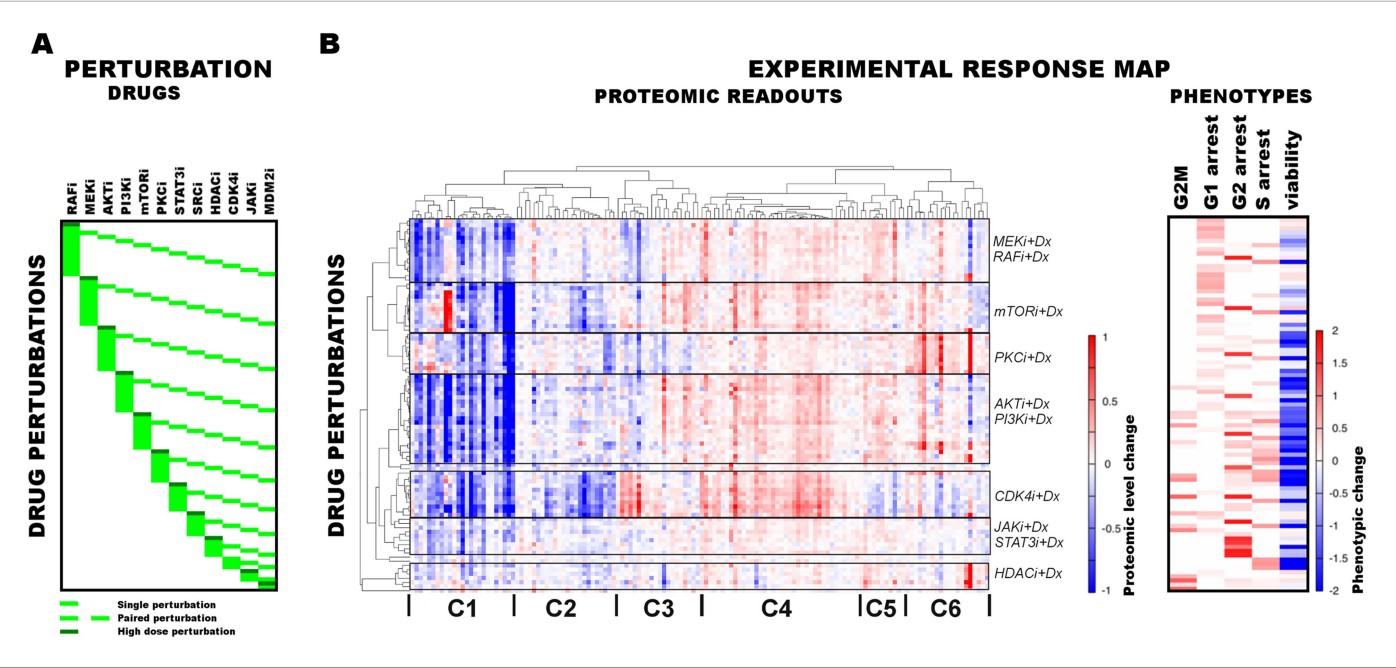

**Figure 2**. Response of melanoma cells to systematic perturbations with targeted agents. (**A**) The combinatorial perturbation matrix. The melanoma cells are perturbed with combinations of targeted drugs (see **Supplementary file 1C** for perturbation conditions). (**B**) The concentration changes in 138 proteomic entities (50 phospho, 88 total protein measurements) (left) and the phenotypic changes (right) in response to drug combinations with respect to the untreated conditions form an experimental 'response map' of the cellular system. The response map reflects the functional relations between signaling proteins and cellular processes. The two-way clustering analysis of the proteomic readouts reveals distinct proteomic response signatures for each targeted drug. Phosphoproteomic response is measured using the RPPA technology. Cell cycle progression and viability response are measured using flow cytometry and resazurin assays, respectively. The cell cycle progression phenotype is quantified based on the percentage of the cells in a cell cycle state in perturbed condition with respect to the unperturbed condition. For the phenotypic readouts, the order of the perturbation conditions is same as in (**A**). The response values are relative to a no drug control and given as $\log_2$(perturbed/unperturbed).

The following figure supplements are available for figure 2:

**Figure supplement 1**. The two-way clustering analysis of the experimental response map.

**Figure supplement 2**. Cluster-guided pathway analysis of the response map.

**Figure supplement 3**. Proteomic response to single agent perturbations.

**Figure supplement 4**. Association models of the response map from partial correlations.

response under 89 perturbation conditions (**Figure 2B,C**). We used reverse phase protein arrays (RPPA) to collect drug response data for 138 proteomic (total- and phospho-protein levels) entities in all conditions (**Tibes et al., 2006**). In parallel, we measured phenotypic responses, including cell viability and cell cycle progression (i.e., G1, S, G2, G2M arrest phenotypes) in all conditions (**Figure 2B**).

## The response map

The high-throughput phenotypic and proteomic profiles form a response map of cells to systematic perturbations (**Figure 2**). The response map provides context-specific experimental information on the associations between multiple system variables (i.e., proteomic entities) and outputs (i.e., phenotypes) under multiple conditions (i.e., perturbations). We demonstrate through hierarchical clustering of the map that each targeted drug induces a distinct proteomic response, and drugs targeting the same pathway lead to overlapping responses in the SkMel-133 cells (**Figure 2B**). Through a clustering-driven pathway analysis, we further show that functionally related proteins (i.e.,

proteins on same or related pathways) respond similarly to targeted agents (*Figure 2C*, *Figure 2—figure supplements 1–3*).

The response map can be described as a set of uncoupled pairwise associations between proteomic and phenotypic entities. We provide an example of an association network model (*Figure 2—figure supplement 4*). The association network was inferred using the experimental response map (*Figure 2*) and graphical Gaussian models, which utilizes partial correlations for network inference (*Schafer and Strimmer, 2005*). Such models, however, are not sufficient for achieving systematic predictions as they do not capture the nonlinear nature of the couplings between the entities and cannot be executed with in silico perturbations. Therefore, we built quantitative models using the experimental response map. The resulting models describe the coupled nature of the interactions between proteins and cellular events, as well as the nonlinear dynamics of cellular responses to drug perturbations.

## Quantitative and predictive network models of signaling

### Network models

Next, we used the experimental response map (*Figure 2*) and the BP-based inference strategy (*Figure 1*) to build quantitative network models of signaling in melanoma. In the models, each node quantifies the relative response of a proteomic or phenotypic entity to perturbations with respect to the basal condition. Consequently, proteomic entities that do not respond to even a single perturbation condition, do not contribute any constraints for inference. We eliminated such entities from the network modeling with a signal-to-noise analysis and included 82 of the 138 proteomic measurements in the modeling (see 'Materials and methods', *Supplementary file 1B*). In addition to the proteomic nodes, the models contained 5 phenotypic nodes and 12 'activity nodes', which represent the 12 drugs and couple the effects of the targeted perturbations to the other nodes in the network. In total, network models contained 99 nodes. BP algorithm generates the probability distribution of edge strengths for every possible interaction between the nodes. The BP-guided decimation algorithm (see 'Materials and methods', *Figure 1—figure supplement 1*) instantiates distinct network model configurations from the probability model (*Montanari et al., 2007*).

The mathematical formulation of the BP-based network inference is suitable for both de novo modeling (i.e., modeling with no prior information) and modeling using prior information (see 'Materials and methods'). Here, we used prior information to infer models with higher accuracy and predictive power compared to de novo models. Using the PERA computational tool, we derived a generic prior information model from Reactome and NCI-Nature PID databases, which were stored in Pathway Commons (*Cerami et al., 2011*). The prior information network contains 154 interactions spanning multiple pathways (*Figure 4—figure supplement 1*). Next, we added a prior prize term to the error model (*Equation 2*) to restrain the search space by favoring the interactions in the prior model. It is critical that the prior information does not overly restrain the inferred models and the algorithm can reject incorrect priors. To address this problem, we inferred network models using the pathway driven and randomly generated prior restraints. The statistical comparison of the networks inferred with actual (i.e., reported in databases) and random prior models indicated that the inference algorithm rejected a significantly higher number of prior interactions when randomly generated priors were used for modeling (*Figure 4—figure supplement 3*). Finally, we integrated the experimental data and prior information to generate 4000 distinct and executable model solutions with low errors (i.e., sets of model parameters, $W_{ij}$, for which the model equations best reproduce the experimental response map) using the BP-based strategy.

## Use of prior information improves the predictive power of models

### Cross validations with and without prior information

We addressed the question whether BP-derived models have predictive power and whether use of prior information introduces further improvement. To assess the predictive power of the network models (i.e., predicting the response to untested perturbations), we performed a leave-k-out cross validation (*Figure 3A*). In each validation calculation, we withheld the response profile to every combination of a particular drug (e.g., RAFi) and all other drugs (leave-11-out cross validation) (*Figure 3B*, *Figure 3—figure supplement 1*). For each cross-validation calculation, this procedure

created a partial training data set that contains response to combinations of 11 drugs and two different doses of a single drug totaling to 78 unique conditions (*Supplementary file 1B,C*).

First, we constructed both de novo (i.e., without any prior information) and prior information guided network models using two partial data sets from which responses to combinations of drugs with RAFi or AKTi were withheld. Next, we predicted the response by executing the models with in silico perturbations that corresponded to the withheld experimental conditions. Finally, we compared the hidden and predicted response data from models generated de novo or with prior information.

## Restraining inference with prior information improves the predictive power of models

The comparison between the predicted and the withheld experimental profiles suggests that the de novo network models have considerable predictive power, and the use of prior information in modeling introduces significant improvement in the prediction quality (*Figure 3B*). Use of prior information increased the cumulative correlation coefficient between predicted and experimental response data from 0.72 to 0.84 and from 0.71 to 0.81 for RAFi and AKTi, respectively. The demonstrated predictive power of the models suggests that the models are suitable for systematically predicting response to perturbation combinations not sampled in the training set and generating testable hypotheses that link external perturbations (e.g., targeted drugs) to cellular response.

## Network models identify context-dependent oncogenic signaling in melanoma

### Network modeling and the average model

We generated quantitative network models with the complete experimental response profile and prior information to investigate oncogenic signaling in melanoma. The resulting network models resemble conventional pathway representations facilitating their comparison with the biological literature (*Le Novere et al., 2009*), but the interaction edges do not necessarily represent physical interactions between connected nodes. Analysis of the ensemble of network model solutions reveals that a set of strong interactions is shared by a majority of the inferred network models. On the other hand, some interactions have nonzero edge strength ($W_{ij} \neq 0$) values only in a fraction of the models (see *Figure 4—figure supplement 2* for analysis of the edge distribution in models). As a first step of detailed analysis and for the purpose of intuitive interpretation, we computed an average network model (*Figure 4A*), which is obtained by averaging the interaction strength ($W_{ij}$) for each node pair, ij, over all individual model solutions. This average network model highlights the interactions with high $W_{ij}$ values if shared by the majority of the distinct solutions. Although the average model cannot be simulated to predict system response, it is particularly useful for qualitative analysis of the inferred signaling interactions.

### Global analysis of average models

The average network model provides a detailed overview of the signaling events in melanoma cells (*Figure 3A*). The average model contains 202 unique interactions (127 activating and 75 inhibitory interactions) between 99 signaling entities. 89 of the 154 interactions in the prior model conform to the experimental data, and therefore, are accepted in the majority of the model solutions by the inference algorithm and included in the average model (*Figure 4—figure supplement 4*). Given that the average model covers interactions from multiple signaling pathways and is more complex than the pathway diagrams presented in most review papers, even the qualitative analysis of the model is highly challenging.

### Network models capture known signaling pathways

In order to simplify the analysis of the average model solution, we fragmented the global network diagram into subnetworks (*Figure 4*). Each subnetwork is a simplified representation of the signaling events in canonical pathways such as those that fall into ERK, PI3K/AKT, and cell cycle pathways (*Figure 4C–E*). The subnetwork diagrams indicate that models recapitulate many known interactions in pathways, which are important in melanoma tumorigenesis (e.g., PI3K/AKT and ERK) and nominate previously unidentified interactions (see *Figure 4* legend). It is, however, not possible to predict the cellular response to untested drug perturbations through qualitative analysis of the inferred

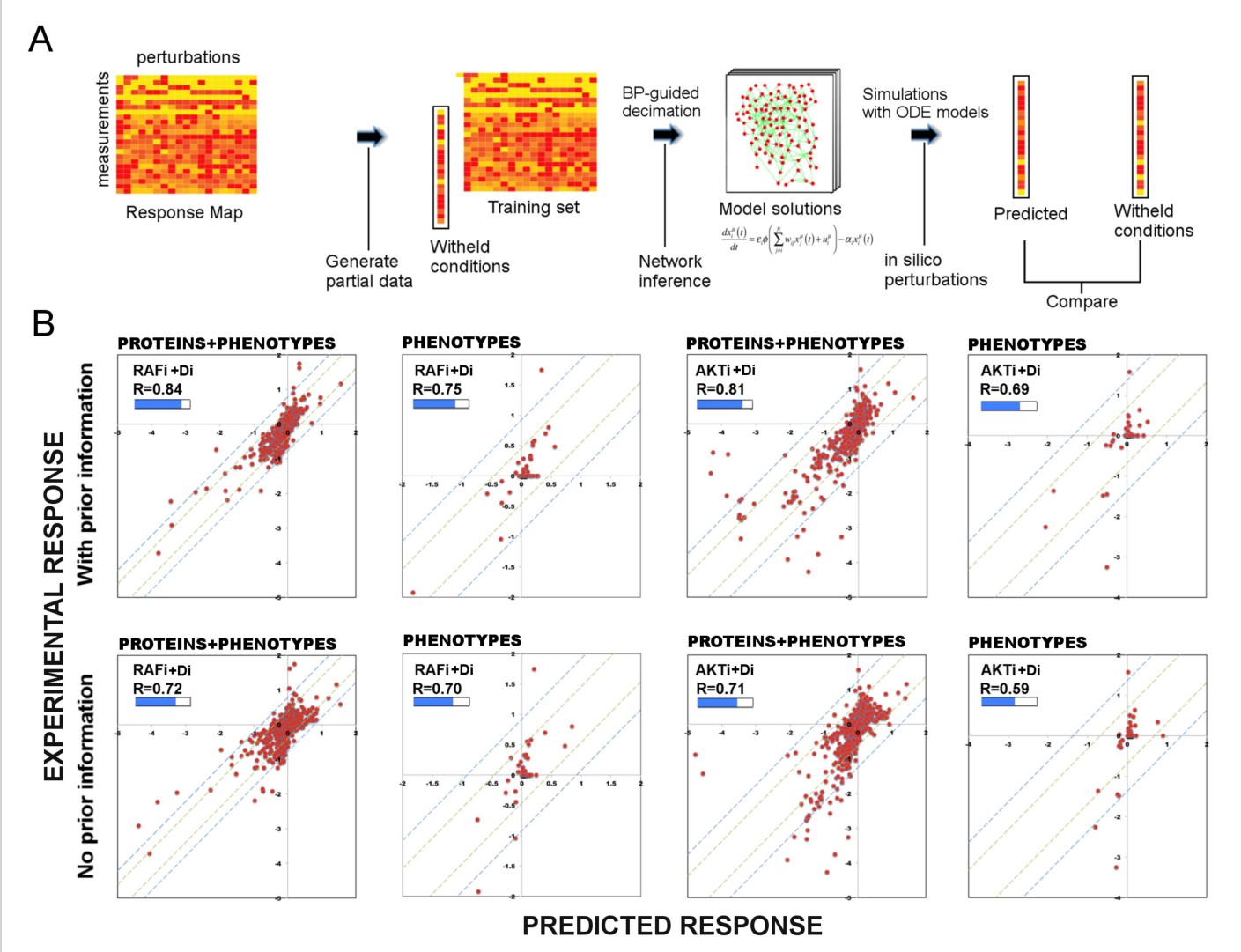

**Figure 3**. Use of prior information increases the predictive power of models. (**A**) To test the predictive power of network models, a leave-11-out cross-validation test is performed. Using the BP-guided decimation algorithm, 4000 network model solutions are inferred in the presence and absence of prior information using the partial response data. Resulting models are executed with in silico perturbations to predict the withheld conditions. Each experimental data point represents the readouts from RPPA and phenotype measurements under the corresponding perturbation conditions. Each predicted data point is obtained by averaging results from simulations with in silico perturbations over 4000 model solutions. The experimental and predicted profiles are compared to demonstrate the power of network models to predict response to combinatorial drug perturbations. (**B**) In all conditions, network inference with prior information leads to a higher cumulative correlation coefficient (R) and significantly improved prediction quality (RAFi $p = 1 \times 10^{-3}$, AKTi $p = 5.7 \times 10^{-3}$, unpaired t-test $H_0$: $\Delta X^{with\_prior} = \Delta X^{w/o\_prior}$, $\Delta X = |X_{exp} - X_{pred}|$) between experimental and predicted responses. Plots on top row: prior information is used for network inference. Plots on bottom row: no prior information is used for network inference. Response to RAFi + {$D_i$} (first and second column) and AKTi + {$D_i$} (third and fourth column) is withheld from the training set and the withheld response is predicted. All responses (phenotypic + proteomic) (first and third column) and only phenotypic responses (second and fourth column) are plotted. {$D_i$} denotes set of all drug perturbations combined with drug of interest. (See *Figure 3—figure supplement 1* for the cross-validation calculations with all other partial data sets and *Supplementary file 1E* for statistical validation of the predictions.)

The following figure supplement is available for figure 3:

**Figure supplement 1**. Predictive power of the perturbation biology models.

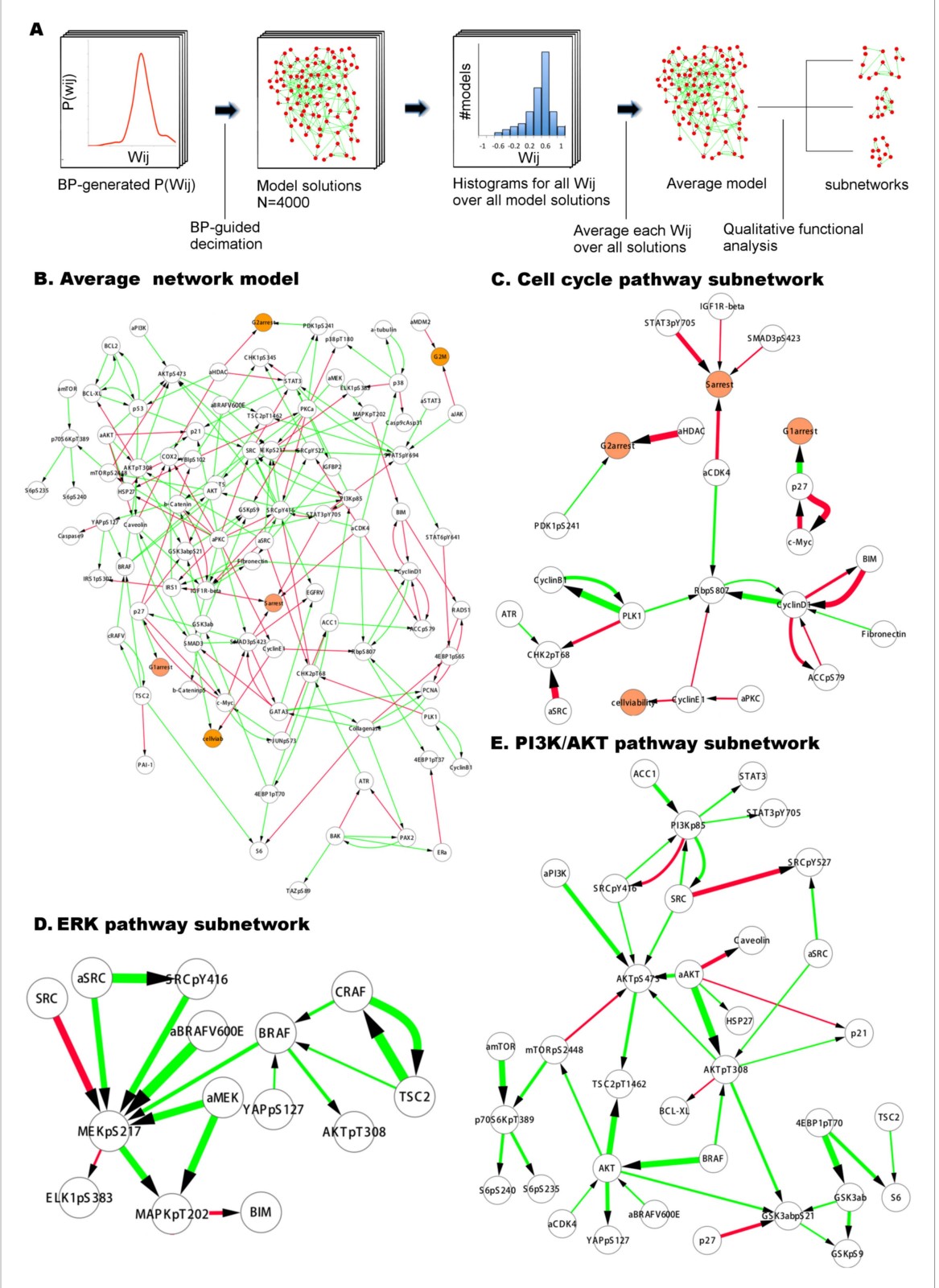

**Figure 4.** Inferred network models capture oncogenic signaling pathways in melanoma. (**A**) The generation of the average model. The set of <Wij> averaged over the Wij in all models provide the average network model. The signaling processes are explained through qualitative analysis of the average model and its functional subnetworks (see **Figure 5** for quantitative analysis). (**B**) The average network model contains proteomic (white) and phenotypic nodes (orange) and the average signaling interactions (<Wij> > 0.2) over the model solutions. The edges between the BRAF, CRAF, TSC2, and AKTpT308

*Figure 4. continued on next page*

*Figure 4. Continued*

represent the cross-pathway interactions between the MAPK and PI3K/AKT pathways (see *Figure 2—figure supplement 2* for analysis of edge distributions in the solution ensemble) (green edges: positive signed, red edges: negative signed interactions). (**C**) Cell cycle signaling subnetwork contains the interactions between the cyclins, CDKs, and other associated molecules (e.g., p27/Kip1). RBpS807 and cyclin D1 are the hub nodes in the subnetwork and connect multiple signaling entities. (**D**) ERK subnetwork. MEKpS217 is the critical hub in this pathway and links upstream BRAF and SRC to downstream effectors such as ERK phosphorylation. (**E**) In the PI3K/AKT subnetwork, the SRC nodes (i.e., phosphorylation, total level, activity) are upstream of PI3K and AKT (total level, AKTpS473 and AKTp308) and the AKT nodes are the major hubs. Downstream of AKT, the pathway branches to mTOR, P70S6K and S6 phosphorylation cascade and the GSK3β phosphorylation events. A negative edge originating from mTORpS2448 and acting on AKTpS473 presumably captures the well-defined negative feedback loop in the AKT pathway (*O'Reilly et al., 2006*). Note that nodes tagged with 'a' (e.g., aBRAFV600E) are activity nodes, which couple drug perturbations to proteomic changes.

The following figure supplements are available for figure 4:

**Figure supplement 1**. The prior model of signaling.

**Figure supplement 2**. Distribution of edges in the solution ensemble.

**Figure supplement 3**. Comparison of random and actual prior information.

**Figure supplement 4**. A subset of the prior information is represented in the average network model.

**Figure supplement 5**. Optimization of network inference cost function parameters.

interactions. We use quantitative simulations with in silico perturbations to both decode the signaling mechanisms and more importantly systematically predict cellular response to combinatorial drug perturbations.

## Combinatorial in silico perturbations generate an expanded proteomic/phenotypic response map

### Model execution with in silico perturbations

Thanks to their ordinary differential equation (ODE)-based mathematical descriptions, the models can be executed to predict cellular responses to novel perturbations (*Nelander et al., 2008*). The systematic predictions from the models go beyond the analysis of few particular edges in the system and capture the collective signaling mechanisms of response to drugs. We execute the parameterized model ODEs (*Equation 1*) with in silico perturbations acting on node (i) as a real numbered $u(i)$ value until all the system variables (i.e., node values, $\{x_i\}$) reach to steady state (*Figure 5A,B*).

### Prediction of phenotypic responses

The simulations expand the size of the response map by three orders of magnitude (i.e., from few thousand experimental response data to millions of predicted responses) (*Figure 5C*, *Figure 5—figure supplement 1*). Once we had the predicted response profiles, we searched for specific perturbations that may induce desired phenotypic changes even when cells are treated with drugs at physiologically relevant doses. For this purpose, we identified the top ranked perturbation conditions in terms of their predicted influence on each phenotype and nominated them for further experimental testing (*Table 1*, *Supplementary file 1F*). Not surprisingly, the execution of models quantitatively recaptured the experimentally observed associations between the drug perturbations and the phenotypic responses. For example, targeting PKC or CDK4 with specific kinase inhibitors lead to reduction of cell viability according to the simulations, which can also be directly observed from the experiments. However, CDK4 and PKC inhibitors substantially reduced SkMel-133 cell viability only at high doses, as in the original perturbation experiments (>2 µM) (*Figure 6—figure supplement 1*). More importantly, the simulations allowed us to identify effective perturbation combinations that cannot be trivially deduced from the experimental data (*Table 1*, *Figure 5D–H*, *Supplementary file 1F*).

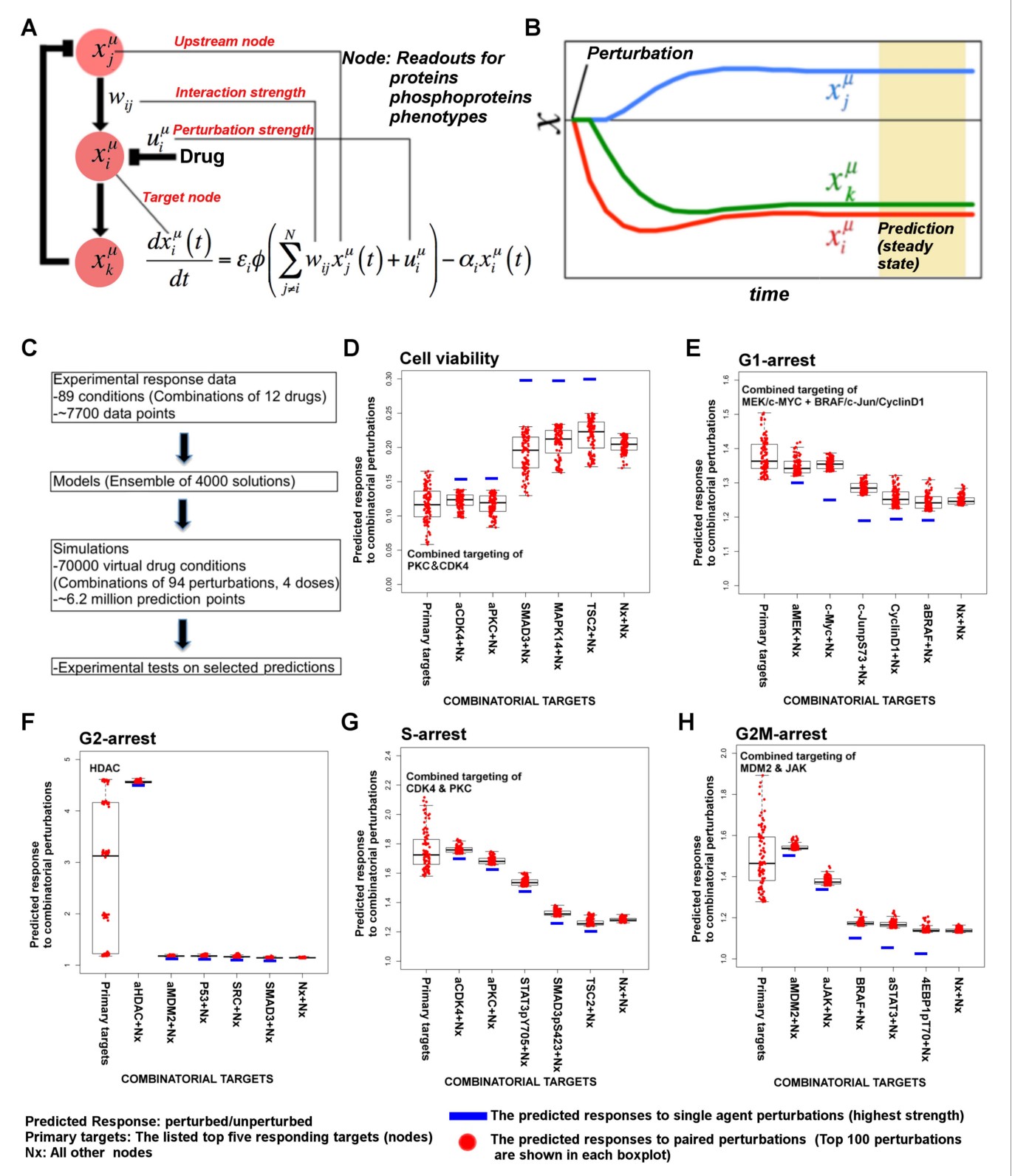

**Figure 5**. Simulations with in silico perturbations provide predictions on system response to novel perturbations. (**A**) The schematic description of network simulations. The system response to paired perturbations is predicted by executing the ODE-based network models with in silico perturbations. In the ODE-based models, {$W_{ij}$} represents the set of interaction strengths and is inferred with the BP-based modeling strategy. The in silico perturbations are

*Figure 5. Continued*

applied as real-valued $u_i^m$ vectors. The time derivative and final concentration of any predicted node is a function of the model parameters, the perturbations, and the values of all the direct and indirect upstream nodes in the models. (**B**) The model equations are executed until all model variables (protein and phenotype responses) reach to steady state. The predicted response values are the averages of simulated values at steady state over 4000 distinct model solutions. (**C**) The simulations expand the response map by three orders of magnitude and generate testable hypotheses. (**D–H**) The predicted phenotypic response to combinatorial in silico perturbations. Each box contains the 100 highest phenotypic responses to paired perturbations. The first box includes the response predictions for combined perturbations on primary targets (e.g., aMEK, c-Myc for G1-arrest. Also see *Table 1* for the definition of the term 'primary target'). The second to sixth boxes include the predicted response for combined targeting of the primary targets with all other nodes (Nx). The last box represents the predicted response data for combination of all nodes except the primaries. For the complete predicted phenotypic response see *Figure 5—figure supplements 1, 2*.

The following figure supplements are available for figure 5:

**Figure supplement 1**. Prediction of phenotypic responses to in silico combinatorial perturbations.

**Figure supplement 2**. The top ten most effective single-agent in silico perturbations for each phenotype.

The execution of the models identified a set of perturbation conditions that are likely to induce a strong phenotypic response. The listed predicted targets (*Table 1*, *Supplementary file 1F*) will serve as a useful resource to nominate intervention strategies to overcome RAFi resistance in melanoma. In particular, in silico perturbations of c-Myc lead to the strongest G1-arrest response when c-Myc is co-targeted with BRAF, MEK, and cyclin D1 (*Figures 5D–H, 6A*). The G1-arrest phenotype is of particular interest given that induction of G1-arrest is a promising strategy for controlling melanoma growth (*Solit et al., 2006*) and various genomic alterations associated with G1/S transition are involved in tumor progression (*Rother and Jones, 2009*). Consistent with the vast literature on c-Myc's role in genesis of many cancers (*Dang, 2012*), predictions indicated that c-Myc linked multiple pathways such as ERK and PI3K/AKT to regulation of cell cycle arrest (*Table 1*; also see *Figure 6—figure supplement 2* for the highly complex c-Myc response to targeted perturbations in the underlying RPPA data). As neither c-Myc nor its direct regulators were inhibited in the perturbation experiments, the predictions from the models were nontrivial and we decided to test these predictions experimentally.

**Table 1**. Phenotypic nodes and predicted responses from simulations with in silico perturbations

| Phenotype | Immediate upstream nodes ($\langle w_{ij} \rangle$ in average model) | Predicted primary target* | Predicted combination partners |
|---|---|---|---|
| Cell viability | SMAD3 (0.49) | aPKC | SMAD3, 4EBP1pT70, TSC2, MAPK14/p38 |
| | cyclin E1 (−0.28) | aCDK4 | |
| G1-arrest | p27/KIP1 (0.83) | c-Myc | Cell cycle (cyclin B1, RBpS807, cyclin D1, PLK1), MAPK pathway (MAPKpT202, MEKpS217, BRAF, aBRAF), AKT, AKTpT308, RAD51, p38/MAPK14, aSRC, YBIpS102, cJUNpS73, SMAD3 |
| | PDK1pS241 (0.20) | aMEK | |
| G2-arrest | aHDAC (−0.77) | aHDAC | Generic response from all nodes |
| S-arrest | STAT3pY705 (−0.66) | aCDK4 | aPKC, SMAD3pS423, STAT3pS705, IGFBP2, cyclin B1, IGF1Rβ, Fibronectin |
| | IGF1Rβ (−0.20) | | |
| | SMAD3pS423 (−0.23) | aPKC | STAT3pS705 |
| | aCDK4 (−0.49) | | |
| G2M | aJAK (−0.24) | aMDM2 | aJAK, aSTAT3, BRAF, 4EBPpT70 |
| | aMDM2 (−0.38) | | |

*A proteomic node is a primary target when substantial phenotypic change is predicted in response to perturbation of the node alone. The phenotypic response is further increased when the primary targets are perturbed in combination with a set of other nodes (i.e., the combination partners). Also See *Figure 5D–H*, *Supplementary file 1D* and *Figure 5—figure supplements 1, 2*.

## Co-targeting c-Myc with MEK or RAF is synergistic in melanoma cells

We predicted through quantitative simulations that melanoma cells were arrested in G1-phase of the cell cycle when c-Myc was targeted alone or in combination with other proteins, particularly BRAF, MEK, and cyclin D1 (*Figure 6A*). We experimentally tested the key prediction from the network models. In order to target c-Myc expression, we treated melanoma cells with the BET bromodomain inhibitor, JQ1, as a single agent and in combination with MEKi (PD-0325901) or RAFi (vemurafenib). JQ1 directly targets bromodomains, especially those of the bromodomain protein BRD4, which marks select genes including *MYC* on mitotic chromatin. Inhibition of the BRD4 bromodomains with JQ1 downregulates *MYC* mRNA transcription and leads to G1 cell cycle arrest in diverse tumor types, such as multiple myeloma (*Delmore et al., 2011*; *Loven et al., 2013*; *Puissant et al., 2013*).

First, we asked whether we could affect c-Myc levels in SkMel-133 cells using JQ1. As measured by Western blot experiments, c-Myc protein expression is reduced in response to JQ1 alone. c-Myc protein levels are further reduced when the cells are treated with a combination of JQ1 and MEKi or RAFi (*Figure 6B*).

To directly test the key prediction from the perturbation biology models, we measured the cell cycle progression response of melanoma cells to JQ1 in combination with the RAF and MEK inhibitors. We observed a strong synergistic interaction between JQ1 and RAFi (*Figure 6C,D*). 51% and 46% of melanoma cells are in G1-stage 24 hr after treatment with JQ1 (500 nM) and RAFi (200 nM), respectively, while 39% of cells are in G1-stage in the absence of any drug. On the other hand, when cells are treated with the combination of JQ1 and RAFi, a drastic increase in the fraction of cells arrested in G1-stage (84%) is observed. The single agent MEKi (50 nM) induces a strong G1-arrest phenotype in SkMel-133 cells (88% G1-stage in MEKi-treated cells vs 39% in nondrug treated cells). The combination of MEKi with JQ1 arrests an even higher fraction of the cells (92%) in the G1-stage (*Figure 6—figure supplement 3*).

Before assessing the effect of JQ1-MEKi/RAFi combination on viability of melanoma cells (SkMel-133), we tested the effect of single agent JQ1 and found that the melanoma cells were considerably sensitive to single agent JQ1 treatment (cell viability IC50 = 200 nM). The sensitivity of SkMel-133 to JQ1 is similar to those of A375 and SkMel-5 lines (RAFi/MEKi sensitive, carrying *BRAFV600E* mutation) to another BRD4 inhibitor, MS417 (*Segura et al., 2013*). The observed sensitivity is also comparable to those of multiple myeloma and MYCN-amplified neuroblastoma cell lines, reported to be potentially JQ1-sensitive tumor types (*Delmore et al., 2011*; *Puissant et al., 2013*), and substantially higher than those of lung adenocarcinoma and MYCN-WT neuroblastoma cell lines (*Lockwood et al., 2012*; *Puissant et al., 2013*).

We tested the effect of combined targeting of c-Myc with MEK or BRAF on cell viability in SkMel-133 cells (*Figure 6E*). Strikingly, when combined with JQ1 (120 nM), cell viability is reduced by 50% with 120 nM of RAFi (PLX4032), whereas the IC50 for single agent RAFi is >1 μM in RAFi-resistant SkMel-133 cells. Similarly, when combined with 5 nM MEKi (PD901), viability of SkMel-133 cells is reduced by 50% with 100 nM of JQ1, an IC50 value, which is close to those of the most sensitive multiple myeloma cell lines (*Delmore et al., 2011*). At higher doses (IC80), JQ1 is synergistic with both MEKi (combination index, $CI_{85} = 0.46$) and RAFi ($CI_{85} = 0.47$) in SkMel-133 cells. At intermediate doses, JQ1 synergizes with RAFi ($CI_{50} = 0.65$) and has near additive interaction with the MEKi ($CI_{50} = 0.85$) (*Figure 6F*). Consistent with the observed synergy at high doses, both JQ1 combinations significantly improve the maximal effect level ($A_{max}$, response to the drugs at highest doses), leading to lower cell viability beyond the levels reached by treatment with any of the agents alone. The observed improvement in $A_{max}$ is particularly important since a subpopulation of cancer cells usually resist treatment even at highest possible drug doses. Treatments with drug combinations, such as those tested here will overcome or delay emergence of drug resistance if they can shrink the size of this resistant subpopulation (i.e., lead to improved $A_{max}$).

## Discussion

We performed a series of systematic perturbation experiments and, measured proteomic and phenotypic responses in each perturbation condition. Using the high-throughput response profiles, we generated network models of signaling in melanoma cells to systematically predict cellular responses to

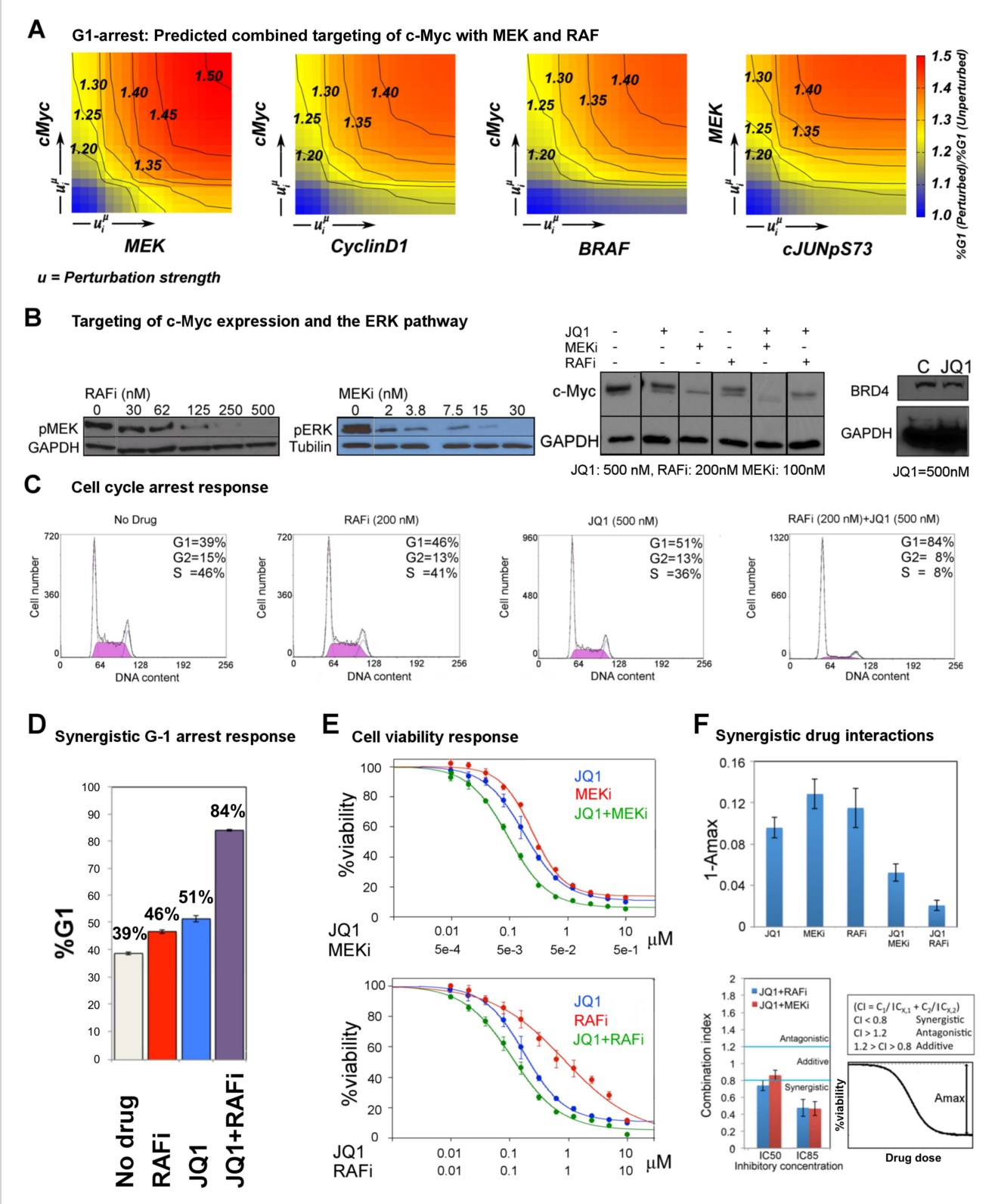

**Figure 6**. The combined targeting of c-Myc with MEK and BRAF leads to synergistic response in melanoma cells. (**A**) The isobolograms of predicted G1-response to combined targeting of c-Myc with MEK, BRAF, cyclin D1, and pJUNpS73. The leftward shift of isocurves implies synergistic interactions between the applied perturbations particularly for co-targeting of c-Myc with MEK or BRAF. **u** denotes strength of in silico perturbations. (**B**) RAFi inhibits MEK phosphorylation at S217 and MEKi inhibits ERK phosphorylation at T202 in a dose-dependent manner (first 2 gels). Western blot shows the level of

*Figure 6. continued on next page*

*Figure 6. Continued*

c-Myc in response to JQ1, MEKi, RAFi, and their combinations (24 hr) (third gel). c-Myc expression is targeted with JQ1 combined with MEKi or RAFi. Direct target of JQ1, BRD4 protein is expressed in both control and 500 nM JQ1-treated cells (fourth gel) (See *Figure 6—figure supplement 4* for uncropped western blot images). (**C**, **D**) The cell cycle progression phenotype in response to JQ1 and RAFi as measured using flow cytometry. 46% and 51% of cells are in G1-stage 24 hr after RAFi and JQ1 treatment, respectively. The combination has a synergistic effect on G1 cell cycle arrest (G1 = 84%). 39% of cells are in G1 when they are not treated with drugs. On panel **D**, error bars in right panel: ±SE in three biological replicates **E**. The drug dose–response curves of cell viability for MEKi + JQ1 (top) and RAFi + JQ1 (bottom). Cell viability is measured using the resazurin assay. Error bars: ±SEM in three biological replicates **F**. The synergistic interactions between JQ1 and RAFi/MEKi. $1 - A_{max}$ is the fraction of cells alive in response to highest drug dose normalized with respect to the nondrug treated condition (top panel). Combination index (CI) quantifies the synergistic interactions between drugs (bottom left). CI is calculated at a given level of inhibition and is a measure of the fractional shift between the combination doses (C1 and C2) and the single agent's inhibitory concentration ($C_{x,1}$, $C_{x,2}$).

The following figure supplements are available for figure 6:

**Figure supplement 1**. The cell viability response to CDK4i and PKCi.

**Figure supplement 2**. Changes in c-Myc level in response to perturbations in SkMel-133 cells as measured with RPPA experiments.

**Figure supplement 3**. Cell cycle arrest response of melanoma cells to JQ1 and MEKi combination.

**Figure supplement 4**. Uncropped versions of the gels presented in *Figure 6B*.

untested drug perturbations. Our modeling algorithm integrates high-throughput drug response profiles and pathway information from signaling databases. The scale and the predictive power of the models were beyond the reach of the previously available network modeling methods, and we made substantial progress in the design of experiments, acquisition of large profiling data sets, and improving a very efficient probabilistic network inference method. Based on the predictions from the data-driven network models, we found that co-targeting MEK or BRAF with c-Myc leads to synergistic responses that render a melanoma cell line more sensitive to inhibition of BRAF. Beyond nomination of effective drug combinations in cell lines, we suggest that the perturbation biology method more generally paves the way for model-driven quantitative cell biology with diverse applications in many fields of biology.

The problem of insensitivity or resistance to targeted therapy is both important and complicated. Oncogenic alterations that affect sensitivity to targeted therapies may pre-exist in the tumor before treatment or emerge sequentially as the result of treatment. Tumors can escape drug treatment through diverse routes and dominant mechanisms of resistance are the exception (such as the T790M mutation in EGFR, which confers resistance to EGFR inhibitors in many patients as well as in the laboratory [*Pao et al., 2005*]), so the identification of effective countermeasures is in general difficult. Typically researchers have identified mechanisms of resistance in individual cases and proposed countermeasures based on the particular mechanism. For example, PTEN loss was associated with resistance to RAFi (in BRAFV600E mutated melanoma) and as a result joint inhibition of BRAF and PI3K was proposed as a combination therapy (*Villanueva et al., 2010*; *Xing et al., 2012*). Based on similar logic, other plausible therapeutic combinations that target particular escape routes in RAFi-resistant melanomas have been put forward, including combinations that target downstream elements of the ERK pathway (e.g., RAFi + MEKi) (*Lito, Pratilas et al., 2012*; *Flaherty et al., 2012a*), or upstream receptor tyrosine kinases (*Prahallad et al., 2012*; *Sun et al., 2014*). For almost all cases, however, tumor cells rapidly adapt and escape through alternate routes after a brief period of response to the combination. Even the FDA-approved RAFi + MEKi combination has only achieved limited improvements in the median progression-free survival in BRAFV600E mutated melanoma patients compared to single agent RAFi treatment (*Long et al., 2014*). A general solution for resistance to the RAFi in BRAF-mutated melanoma, in particular, is not yet in sight.

Aiming to develop a more general strategy for preventing the emergence of drug resistance, we have taken a more systematic approach to the discovery of effective combination therapy and cast a wider net by perturbing cancer cells with an entire set of targeted inhibitors and combinations of inhibitors and observing the proteomic signaling response using hundreds of antibodies covering diverse pathways. We expected to generate plausible predictions for a set of effective combinations

and to be able to rank their relative importance, to guide validation experiments. The perturbation biology approach used here identified a highly ranked co-target of RAF, downstream of the ERK pathway, the oncogene c-Myc. To test our predictions, we targeted c-Myc using the epigenetic drug JQ1, a BRD4 inhibitor that negatively affects c-Myc transcription (*Delmore et al., 2011*). We treated SkMel-133 cells with combinations of JQ1 and MEKi or RAFi and showed that both combinations synergistically induced G1-arrest and decreased cell viability.

A key criterion of pharmacological efficacy is the ability of a drug to induce cellular responses at doses sufficient to inhibit its immediate molecular targets. In SkMel-133 cells, however, the cell viability IC50s for RAFi and MEKi are at least one order of magnitude higher than the doses required to reduce the phosphorylation of immediate downstream targets (phospho-MEK and phospho-ERK) by 50% (*Figure 6B*, *Table 2*), and hence, the cells are considered to be relatively resistant to both drugs. When combined with JQ1, the concentrations of RAFi/MEKi that are sufficient to reduce cell viability by 50% are in the range of MEK and ERK phosphorylation IC50s (*Table 2*). This is a direct result of the strong and synergistic increase in G1-arrest phenotype in response to the drug combinations (*Figure 6D*). Thus, the JQ1-MEKi/RAFi combinations shift the required doses to induce phenotypic responses close to the doses required for inhibition of the ERK pathway activity. Interestingly, in an independent study in hematopoietic cancer, the combination of JQ1 affecting the c-Myc levels and a FLT3 tyrosine kinase inhibitor (TKI) was synergistic in overcoming resistance to the FLT3 TKI in acute myeloid leukemia (AML) cells (*Fiskus et al., 2014*). As far as we know, our study is the first demonstration of a combined inhibition of a BET bromodomain (BET-BRD) protein and a protein kinase molecule to overcome drug resistance in solid tumors (*Filippakopoulos and Knapp, 2014*).

To interpret these positive results, we note that c-Myc is a multifunctional transcription factor with a central role in cell cycle progression and receives inputs from diverse signaling pathways, including ERK, PI3K/AKT, Wnt, and TGF-β signaling (*Dang, 2012*). We hypothesize that, beyond the efficacy of co-targeting BRAF kinase and c-Myc, there may be a particular advantage in blocking proliferative signals from various oncogenic pathways that converge on the downstream effector c-Myc. In other words, c-Myc may be a particular 'point of vulnerability' in RAFi drug-resistant cells (*Figure 7*) and this may be reflected in the high rank of c-Myc as a co-target to induce G1-arrest phenotype in the data-driven perturbation biology network models.

Although the experiments here are limited to the SkMel-133 melanoma cell line, we are tempted to put forward a more general hypothesis. We suggest that synergistic co-targeting of a key genomic aberration (e.g., BRAFV600E) and a downstream point of convergence of signaling (e.g., c-Myc) may be a more effective way of overcoming the resistance than co-targeting another upstream pathway (*Figure 7*). While inhibition of the oncogenic activation by the first drug can be compensated by alteration in a number of bypass pathways, it is plausible that most if not all signaling events have to converge on similar, pro-proliferative consequences—in this example via the downstream transcriptional activator c-Myc. Inhibition of a downstream signaling event may therefore be a generally more effective co-targeting strategy for an entire set of alternative upstream bypass mechanisms of resistance. The key proviso, of course, is absence of toxicity, which may be more likely for co-targeting the upstream oncogenic activator (not active in normal cells, e.g., BRAV600E) in combination with a downstream effector (functional in both normal and tumor cells, e.g., c-Myc [*Soucek et al., 2008*]) at optimized drug concentrations. In other words, the upstream–downstream combination of targets may have the dual advantage of generality in overcoming resistance and avoidance of toxicity. To generally test this hypothesis, one would have

**Table 2**. Drug resistance is overcome as the IC50 (cell viability) approaches the IC50 (target phosphorylation)

|  | JQ1 | MEKi | RAFi |
|---|---|---|---|
| IC50 (target phosphorylation) | – | 3 nM* | 65 nM* |
| IC50 (cell viability) single agent | 200 nM | 15 nM | >1 µM |
| IC50 (cell viability) combined with JQ1 | – | 5 nM† | 120 nM† |

*MEKi targets pERK and RAFi targets pMEK. Phopho-IC50s are quantified from gels (*Figure 6B*).
†IC50s for viability and phosphorylation are in same order.

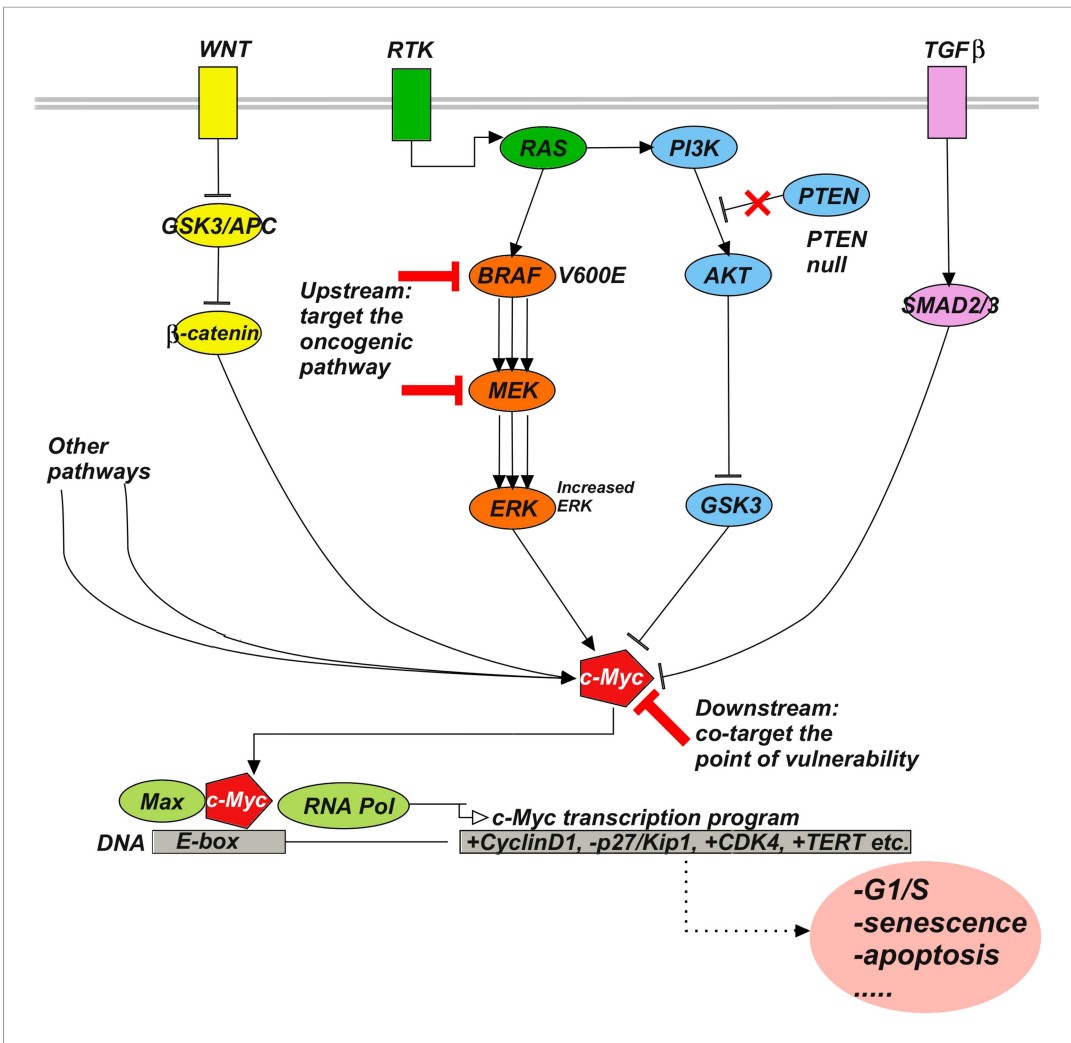

**Figure 7**. Upstream–downstream combination therapy. Co-targeting a specific genomic upstream alteration (here, a BRAFV600E mutant) and a general downstream point of vulnerability (here, c-Myc) may be an effective strategy to overcome drug resistance, as it has the potential dual advantage of specificity and generality. The specificity of targeting an upstream oncogenic activator such as BRAFV600E reduces the likelihood of toxicity because normal cells depend much less or not at all on the activator. The generality of targeting a downstream signaling molecule such as c-Myc increases the likelihood of blocking diverse bypass mechanisms. Signaling pathways such as Wnt, RTK, TGFβ, and various others influence c-Myc stability, expression, and activity. For example, ERK phosphorylates c-Myc at S62 to stabilize the molecule, while phosphorylation at T58 by GSK3 leads to ubiquitylation and degradation. β-catenin and SMAD3 act as transcriptional activator and repressor of c-Myc, respectively. RAF-inhibitor resistance can, for example, be mediated by the activities of the Wnt (*Anastas et al., 2014*), TGFβ (*Sun et al., 2014*), or RTK (*Villanueva et al., 2010*; *Xing et al., 2012*) pathways.

to go beyond the scope of this work and perform co-targeting experiments in other melanoma cell lines, for which the mechanism of resistance is known, or more generally, in other cell lines or mouse models of resistance to targeted therapeutics.

In the current study, we have only tested the JQ1 and RAF-inhibitor combinations in vitro. Future in vivo studies will be important for identifying the optimal BET-BRD inhibitor for combination therapy in melanoma and other tumors. This may be clinically relevant, as BET-BRD inhibitors hold substantial clinical potential for treatment of both solid and hematological tumors. Indeed, the observation that BET-BRD inhibitors (e.g., JQ1) effectively block cellular growth through transcriptional repression of c-Myc in mouse models has created the rationale for clinical trials in c-Myc-dependent cancer types (*Mertz et al., 2011*). Unfortunately, the widely used investigational BET-BRD inhibitor JQ1 has been

reported to have poor in vivo efficacy with low metabolic stability and bio-solubility, but fortunately, other BET-BRD inhibitors with more promising pharmacokinetic properties have been and are being developed. Specifically, JQ1 is reported to have a maximum blood concentration of 0.84 μM in mouse models, whereas the compounds I-BET-762 and I-BET-151 reach maximum blood concentrations of 26 μM and 82 μM, respectively (*Dawson et al., 2011*). In general, triazolobenzodiazepine (e.g., I-BET-762 and I-BET-151 compounds) and triazolothienodiazepine (e.g., OTX015, TEN-010, CPI-0610 compounds) BET-BRD inhibitors are reported to be stable in liver microsomes and have up to 30% bioavailability. These compounds may therefore be more suitable for in vivo applications (*Filippakopoulos and Knapp, 2014*). Several clinical trials of BET-BRD inhibitors with desirable pharmacokinetic properties have been initiated in multiple cancer types, including trials with OTX015 for the treatment of acute leukemia (ClinicalTrials.gov identifier: NCT01713582), CPI-0610 for progressive lymphoma (ClinicalTrials.gov identifier: NCT01949883), I-BET-762 and TEN-010 for NUT midline carcinoma (ClinicalTrials.gov identifier: NCT01587703, NCT01987362), and a dose escalation and cohort expansion study for TEN-010 in patients with AML and myelodysplastic syndrome (ClinicalTrials.gov identifier: NCT02308761). The trials with BET-BRD inhibitors in different cancer sub-types suggest use of such compounds either as single agents or in combinations may be a plausible strategy for treatment of other cancer types, including RAF-inhibitor resistant melanoma.

The model-based predictions provide comprehensive and testable hypotheses on complex signaling mechanisms relevant to the development of novel therapies. More comprehensive experimental data and improvements in signaling databases are likely to lead to network models with even higher predictive power. Interpreting the cell line-specific predictions on the background of genomic profiling of tumor samples from patients may guide therapeutic choices for tumors with similar genomic backgrounds. To assist in these efforts, we have developed genomics methods to classify tumors based on patterns of functional oncogenic alterations and to identify cell lines that most closely resemble tumors (*Ciriello et al., 2013*; *Domcke, Sinha, et al., 2013*). By integrating network models, genomics, and pathway analysis, one can expect to generate whole-cell models of signaling and drug response in mammalian cells with potential applications to personalized medicine and genomically informed clinical trials.

## Website

Website for the perturbation biology method: http://www.sanderlab.org/pertbio/. The website stores the source code for BP-decimation algorithm, input scripts, and data files for running the BP-guided decimation code, perturbation response data, downloadable files for the executable model solutions, and the complete simulation results. A detailed description of the perturbation biology method is also provided in the webpage.

# Materials and methods

## Computational methods

### Automated extraction of prior information from signaling databases

Pathway Extraction and Reduction Algorithm (PERA) was developed to automatically extract prior information from multiple signaling databases and generate a prior information network. The input to PERA is a list of (phospho) proteins identified by their HGNC symbols (e.g., *AKT1*), phosphorylation sites (e.g., pS473), and their molecular status (i.e., activating or inhibitory phosphorylation, total concentration). The output of PERA is a set of directed interactions between signaling molecules represented in a Simple Interaction Format (SIF). The PERA software is available at http://bit.ly/bp_prior as a free software under LGPL 3.0.

### Requirements for prior information

There is a growing number of signaling databases (*Bader et al., 2006*) that capture pathway information in high detail. Compared to the interaction networks with relatively low level of detail, pathway databases contain information about the phosphorylation states of proteins and interactions specific to these states. Although this corpus is highly valuable for phospho-proteomic analysis, it remains mostly untapped. To utilize this knowledge, one needs algorithms to map the experimentally

measured proteins and phosphoproteins to their pathway database equivalents, find connections between them, and reduce the outcome to a format that can be used for inference and analysis.

## The algorithm

We developed the algorithm and the software tool, PERA to automatically extract prior information from multiple signaling databases in the BioPAX (*Demir et al., 2010*) format and generate a prior information network. PERA takes a list of (phospho)proteins identified by their HGNC symbols (e.g., AKT1), phosphorylation sites (e.g., pS473), and their molecular status (i.e., activating or inhibitory phosphorylation, total concentration) as input and then finds directed signaling paths between these entities. These paths are then reduced to directed interactions between signaling molecules represented in a SIF.

## The PERA framework

The prior information is extracted from components of the Pathway Commons 2 database in three steps: first, using the paths-between graph query algorithm (*Dogrusoz et al., 2009*), PERA generates a subgraph (i.e., graph-of-interest) of Pathway Commons, which contains all the input proteins and all known connections within their first neighborhoods. Second, using the phosphorylation and activity state information, input entities are mapped to the corresponding protein states in the graph-of-interest. During this mapping step, protein states that do not match with either the corresponding annotation for phosphorylation or activity state are filtered out. Phosphorylation site mismatches up to six residues are tolerated during the filtering step to account for phosphorylation site ambiguities due to either database curation errors or cross-organism annotations. Third, paths that result in the addition or removal of a phopho-group at a phosphorylation site are extracted and mapped to phosphoprotein nodes. For total protein nodes, all nonphosphoprotein specific, directed signaling paths are included. For this application, the maximum allowable graph query distance limit was set to 1 and only the Reactome (*Matthews et al., 2009*) and NCI-Nature PID pathway (*Schaefer et al., 2009*) data resources were used. Although we limited ourselves to short path distances and two pathway databases that we were most familiar with, PERA can be applied to extract information from any pathway database that exports to BioPAX and can be configured for searching paths of arbitrary length.

## Key considerations for extracting pathway information

Two major semantic issues need to be taken into account when extracting state-specific pathway information from databases. First semantic issue stems from the ambiguities in mapping proteins with multiple phosphorylation sites. When a protein with multiple phosphorylation sites is experimentally profiled with an antibody, which recognizes only a single phosphorylation site, the antibody will actually bind to a heterogeneous mixture of phospho-states provided that the epitope is phosphorylated (e.g., anti-AKTpS473 Ab may bind to both AKTpS473 and AKTpS473_pT308 but not to AKTpT308). For proteins with multiple observed phosphorylation sites, this might lead to semantic conflicts since a double phosphorylated node should be mapped to both observations (i.e., single and double-phosphorylated states). We included an optional 'strict' mapping scheme to map only the phosphoproteins that exactly match the observed node—in our case always single phosphoproteins (e.g., the epitope of anti-AKTpS473 Ab is mapped to AKTpS473 but not to AKTpS473_pT308). Since our extraction algorithm is much more tolerant of missing interactions compared to false interactions, we opted to use this flag for this application.

The second semantic issue stems from the fact that pathway databases are often curated from multiple independent studies, spanning multiple cellular states, cell and tissue types, and even multiple model organisms. As a result, they are a superimposition of possible interactions over a wide range of spatiotemporal and genetic contexts. On the other hand, databases cover only a subset of all possible contexts. In our case, we expect only a subset of the interactions in the pathway databases to be active in our cell lines and cover only a subset of the interactions that we observe. This observation necessitates incorporating prior information as 'soft' restraints for network inference. For this purpose, we devised a modified cost function, which includes a term for prior information (See below. Also see [*Molinelli, Korkut, Wang et al., 2013*] and [*Miller et al., 2013*]).

## Mathematical description of network models

The network models represent the time behavior of the cellular system in a set of perturbation conditions as a series of coupled nonlinear ODEs (*Nelander et al., 2008*).

Equation 1. Network model ODEs

$$\frac{dx_i^\mu(t)}{dt} = \varepsilon_i \phi \left( \sum_{j \neq i}^{N} w_{ij} x_j^\mu(t) + u_i^\mu \right) - \alpha_i x_i^\mu(t), \qquad (1a)$$

$$\phi(z) = \tanh(z). \qquad (1b)$$

In the network models, each node represents the quantitative change of a biological variable, $x_i^\mu$ ((phospho)protein level and phenotypic change) in the perturbed condition, $\mu$, relative to the unperturbed condition. $W_{ij}$ quantifies the edge strength, which is the impact of upstream node j on the time derivative of downstream node i. We assign semi-discrete values to each $W_{ij}$, $\mathbf{W} = \{w_{ij}, \forall w_{ij} \in \{-1, -0.8, \ldots, 0.8, 1\}\}$. $\alpha_i$ constant is the tendency of the system to return to the initial state, and $\varepsilon_i$ constant defines the dynamic range of each variable $i$. The transfer function, $\Phi(x)$ ensures that each variable has a sigmoidal temporal behavior.

## Modified cost function for network inference with prior information

We have quantified the cost of a model solution by an objective cost function C(W). The network configurations with low cost represent the experimental data more accurately. Here, we have incorporated an additional prior information term to the cost function to construct models with improved predictive power. The newly introduced term in the cost function accounts for the prize introduced when the inferred $w_{ij}$ is consistent with the prior information.

In order to generate predictive network models of signaling, we have quantified the cost function C (W) to penalize (a) discrepancies between predicted ($x_i^\mu$) and experimental ($x_i^{\mu*}$) values of the system variables at a time point $\{t_l\}$ in condition μ, and (b) the number of nonzero interactions with an L0 norm (*Nelander et al., 2008*; *Molinelli, Korkut, Wang et al., 2013*). Here, we have also incorporated prior information to construct network models with higher predictive power even if the models are constructed with limited experimental data. The newly introduced third term in the cost function accounts for the prize introduced when the inferred $w_{ij}$ conforms to the prior information ($w_{ij}^{prior}$). The modified cost function with prior information term is formulated as in *Equation 2*.

Equation 2. Modified cost function

$$C(W) = C^{SSE}(W) + C^{complexity}(W) + C^{prior}(W), \qquad (2a)$$

$$C(W) = \beta \sum_{l}^{L} \sum_{i}^{N} \sum_{\mu}^{M} \left( x_i^\mu(t_l) - x_i^{\mu*}(t_l) \right)^2 + \lambda \sum_{i}^{N} \sum_{j \neq i}^{N} \delta(w_{ij}) + \sum_{i}^{N} \sum_{j \neq i}^{N} \eta(w_{ij}), \qquad (2b)$$

$$\begin{aligned} \delta(w_{ij}) &= 1 \ if \ w_{ij} \neq 0 \\ \delta(w_{ij}) &= 0 \ if \ w_{ij} = 0. \end{aligned} \qquad (2c)$$

In *Equation 2*, the first term accounts for the discrepancies between predicted ($x_i^\mu$) and experimental ($x_i^{\mu*}$) values of the system variables at a time point $\{t_l\}$ in condition $\mu$, for a particular network configuration ($\mathbf{W} = \{w_{ij}\}$). The second term is the complexity factor with an L0 norm, which reduces the number of nonzero interactions in a network configuration and ensures that resulting network models are sparse. The final term, η(**W**) is the cumulative prior prize function for **W** = $\{w_{ij}\}$. η($w_{ij}$ = ω) has a negative real value and reduces the overall model cost if the inferred ω conforms to the prior information. Here, we will formulate the newly introduced prior information term in the modified cost function.

### Generalized prior information prize function

In order to derive the prior information prize (PIP) function, we first assume the value of prior $w_{ij}$ is normally distributed (truncated by boundaries of $w_{ij}$) around an expected prior value, E[$w_{ij}^{prior}$] with a standard deviation, $\sigma_{ij}^{prior}$. The prior probability model for each interaction in the prior network is formulized using the normal distribution probability density function around E[$w_{ij}^{prior}$] as in *Equation 3*.

Equation 3. Probability model of prior observations

$$P^{prior}(w_{ij}) = \frac{1}{\sigma^{prior}\sqrt{2\pi}}\exp\left(-\frac{\left(w_{ij} - E\left[w_{ij}^{prior}\right]\right)^2}{2(\sigma^{prior})^2}\right). \qquad (3)$$

The *Equation 3* implies that the probability of inferring a particular $w_{ij}$ value and consequently, the prize introduced to the cost function increase as $w_{ij}$ approaches $E[w_{ij}^{prior}]$. We calculate the prize for the fit between the inferred $w_{ij}$ and the $E[w_{ij}^{prior}]$ with an inverse Boltzmann conversion of the probability model (*Jaynes, 1957*).

Equation 4. Error model of individual prior observations

$$\eta(w_{ij}) = -\kappa_{ij}\ln\left(P^{prior}(w_{ij})\right), \qquad (4a)$$

$$\eta(w_{ij}) = -\kappa_{ij}\ln\left(\frac{1}{\sigma_{ij}^{prior}\sqrt{2\pi}}\exp\left(-\frac{\left(w_{ij} - E\left[w_{ij}^{prior}\right]\right)^2}{2\left(\sigma_{ij}^{prior}\right)^2}\right)\right), \qquad (4b)$$

$$\eta(w_{ij}) = -\kappa_{ij}\ln\left(\frac{1}{\sigma_{ij}^{prior}\sqrt{2\pi}}\right) + \kappa_{ij}\left(\frac{\left(w_{ij} - E\left[w_{ij}^{prior}\right]\right)^2}{2\left(\sigma_{ij}^{prior}\right)^2}\right). \qquad (4c)$$

The first term introduces the maximum possible prize for each interaction that is represented in the prior information network. Second term penalizes the discrepancies between the $w_{ij}$ and the $E[w_{ij}^{prior}]$. The penalty for the discrepancy is zero when $w_{ij} = E[w_{ij}^{prior}]$ and the prior prize, $\eta(w_{ij})$ assumes the highest possible value. $\kappa_{ij}(\ln(1/(\sigma_{ij}^{prior}\sqrt{2\pi})))$ is a constant analogous to the inverse temperature in statistical physics. The constant $\kappa_{ij}$ and the resulting $\eta(w_{ij})$ are zero if the interaction between nodes $i$ and $j$ is not included in the prior network model. By summing over all $i$ and $j$, the cumulative prior prize for all $W = \{w_{ij}\}$ becomes *Equation 5*.

Equation 5. Cumulative error model of prior observations

$$\eta(W) = \sum_{i}^{N}\sum_{j}^{N}\eta(w_{ij}), \qquad (5a)$$

$$\eta(w_{ij}) = \sum_{i}^{N}\sum_{j}^{N}\left(-\kappa_{ij}\ln\left(\frac{1}{\sigma_{ij}^{prior}\sqrt{2\pi}}\right) + \kappa_{ij}\left(\frac{\left(w_{ij} - E\left[w_{ij}^{prior}\right]\right)^2}{2\left(\sigma_{ij}^{prior}\right)^2}\right)\right). \qquad (5b)$$

Even though the current PIP function is derived with the normal distribution assumption, a variety of alternative distributions such as a bimodal distribution can be fitted in *Equation 3*. Such efforts will lead to a variety of custom-designed PIP functions when necessary.

**Estimation of $E[w_{ij}^{prior}]$ and $\sigma_{ij}^{prior}$**

An important requirement for the implementation of *Equation 5* is the accurate estimation of $E[w_{ij}^{prior}]$ and $\sigma_{ij}^{prior}$. Here, we propose two estimation strategies that are alternative to systematic extraction of priors from databases.

**Priors from experiments**

Our first strategy relies on use of biochemical measurements with multiple biological replicates to quantify the influence of node j on node i ($w_{ij}$). The $E(w_{ij}^{prior})$ and $\sigma_{ij}^{prior}$ can be estimated from the mean value ($<w_{ij}^{prior}>$) and standard deviation ($\sigma_{ij}$) of the readouts from biological replicates in such

measurements. Depending on the nature of the interaction under study, a variety of different biochemical methods can be utilized. These methods include but are not limited to enzyme activity assays, quantification of protein–protein interactions (*Selbach and Mann, 2006*), or additional RPPA readouts under specific perturbation conditions.

### Priors from network models

As we build network models of signaling in diverse biological contexts using perturbation data and inference algorithms, a vast amount of information on signaling properties will become available. Consequently, we will be able to estimate the $E(w_{ij}^{prior})$ and $\sigma_{ij}^{prior}$ for newly inferred models from available network models. Previously inferred $\{w_{ij}\}$ values and standard deviation in BP-generated probability distributions will serve as a basis to estimate the $E(w_{ij}^{prior})$ and $\sigma_{ij}^{prior}$, respectively.

### The simplified PIP function

The current form of the prior information is limited to a set of binary interactions due to the qualitative nature of the databases, from which we extract the information. Therefore, we used a simplified PIP function to introduce the prior information restraints to the inference scheme. The prior information stored in the databases may correspond to activating ($w_{ij} > 0$), inhibitory ($w_{ij} < 0$), or generic ($w_{ij} \neq 0$) interactions. In such situations, $w_{ij}^{prior}$ has a uniform distribution within the defined boundary. Thus, the $E[w_{ij}^{prior}]$ can be set as equal to any value within the interval, $[w_{ij}^{min-prior}, w_{ij}^{max-prior}]$. For example, a prior for a positive interaction between nodes i and j is represented by an interval of $(0, w_{ij}^{max})$, where $w_{ij}^{max}$ is the maximum allowed edge strength in the inference scheme. Consequently, the squared distance error in *Equation 4* vanishes and the prior prize assumes a fixed value over the defined prior interval.

Equation 6. Simplified error model of prior information

$$E\left[w_{ij}^{prior}\right] = w_{ij}, \quad \forall w_{ij} = \left\{w \in R \,\middle|\, w_{ij}^{min-prior} \leq w \leq w_{ij}^{max-prior}\right\}, \forall\, i, j \in N, \quad (6a)$$

$$\kappa_{ij}\left(\frac{\left(w_{ij} - E\left[w_{ij}^{prior}\right]\right)^2}{2\left(\sigma_{ij}prior\right)^2}\right) = 0, \quad (6b)$$

$$\eta(w) = \sum_{i}^{N}\sum_{j}^{N}\left(-\kappa_{ij}\, ln\left(\frac{1}{\sigma_{ij}^{prior}\sqrt{2\pi}}\right)\right). \quad (6c)$$

The binary nature of the interactions also implies a generic $\kappa$ value ($\kappa = \kappa_{ij}(ln(1/(\sigma_{ij}^{prior}\sqrt{2\pi})))$, $\kappa_{ij} > 0, \forall\, i, j \in N$) for all interactions represented in prior information network and $\kappa = 0$ for all other situations. The resulting prior information term in the cost function is a step function as shown in the main text methods section.

Equation 7. Simplified error model of individual prior observations

$$\eta\left(w_{ij} \neq 0\right) = -\kappa \text{ and } \eta\left(w_{ij} = 0\right) = 0 \text{ Generic prior information } \left(w_{ij} \neq 0\right)$$

$$\eta\left(w_{ij} > 0\right) = -\kappa \text{ and } \eta\left(w_{ij} \leq 0\right) = 0 \text{ Prior information for an activating interaction}$$
$$\left(w_{ij} > 0\right)$$

$$\eta\left(w_{ij} < 0\right) = -\kappa \text{ and } \eta\left(w_{ij} \geq 0\right) = 0 \text{ Prior information for an inhibitory interaction}$$
$$\left(w_{ij} < 0\right)$$

$$\eta\left(w_{ij}\right) = 0 \text{ No Prior exists for } w_{ij}.$$

$$(7)$$

For each interaction in the prior information network, $\eta(w_{ij})$ has a negative value. Thus, the prior information introduces a prize in the overall cost function. The formulation and implementation into

the BP equations create a soft-restraint for prior information extracted from signaling databases. In our probabilistic inference framework, a prior information is accepted as an edge ($w_{ij} \neq 0$) only if the prize introduced can override any discrepancy between predicted ($x_i^\mu$) and experimental ($x_i^{\mu*}$) values of the system variables. Thus, the context-specific character of inferred models is preserved when the cost function in *Equation 2* is used.

## Network model construction and response prediction

Network models are constructed with a two-step strategy. The method is based on first calculating probability distributions for each possible interaction at steady state with the BP algorithm and then computing distinct solutions by sampling the probability distributions. We described the theoretical formulation, the underlying assumptions and simplification steps of the BP algorithm for inferring network models of signaling elsewhere (*Molinelli, Korkut, Wang et al., 2013*). The network models include 82 proteomic, 5 phenotypic, and 12 activity nodes. Activity nodes couple the effect of drug perturbations to the overall network models (See Molinelli, Korkut, Wang et al. for quantification of activity nodes).

## Belief propagation

Belief propagation (BP) algorithm iteratively approximates the probability distributions of individual parameters. The iterative algorithm is initiated with a set of random probability distributions. In each iteration step, individual model parameters are updated (local updates) based on the approximate knowledge of other parameters, experimental constraints, and prior information (global information). In the next iteration, the updated local information becomes part of the global information and another local update is executed on a different model parameter. The successive iterations continue over different individual parameters until the updated probability distributions converge to stable distributions. The iterations between the local updates and the global information create an optimization scheme that $\mathbf{W} = \{w_{ij}\}$ is inferred given a probability model. Explicitly, the following cavity update equations are iteratively calculated until convergence.

## Equation 8. BP update equations

$$P^\mu(w_{ij}) = \frac{1}{Z_{ij}} e^{-\lambda\delta(w_{ij})} \cdot e^{-\eta(w_{ij})} \prod_{\upsilon \neq \mu}^{M} \rho^\upsilon(w_{ij}) \forall j \neq k, \tag{8a}$$

$$\rho^\mu(w_{ik} = \omega) = \frac{1}{Z_{ik}} \sum_{w_{ij}, j \neq k} e^{-\beta(x_i^\mu - x_i^{\mu*})^2} \prod_{j, j \neq k}^{N} P^\mu(w_{ij}). \tag{8b}$$

In *Equation 8a*, $P^\mu(w_{ij})$ approximates the mean field of the parameters with a sparsity constraint ($\lambda\delta(w_{ij})$) and a bias from prior information restraints ($\eta(w_{ij})$). In *Equation 8b*, $\rho^\mu(w_{ij} = \omega)$ is a mean field derived change to the probability distribution of the locally optimized parameter, towards minimizing the model error ($C^{SSE}(W)$). (See *Molinelli, Korkut, Wang et al. (2013)* for derivation and implementation of BP equations).

**Gaussian integration of cavity parameter update**
Information from a sufficiently large set of noncavity constraints and parameters is integrated to update the cavity parameters. As a corollary to the central limit theorem, we can assume that the mean-field effect of noncavity parameters on the cavity update follows a Gaussian distribution. The mean and variance of the Gaussian distribution are the mean values and variances of the global distributions given by individual distributions of noncavity $P^\mu(w_{ij}) \forall j \neq k$. Thus, we replace the sum over multivariate configurations of all noncavity parameters with a single Gaussian integration over the mean-field of noncavity configurations.

## Equation 9. Gaussian approximation to the cavity parameter update

$$\rho^\mu(w_{ik} = \omega) = \frac{1}{Z_{ik}} \frac{1}{\sqrt{2\pi\Delta_k^\mu}} \int_{-\infty}^{\infty} e^{-\beta\left(x_i^{\mu*} - \phi\left(s_k^\mu + w_{ik} x_k^{\mu*}\right)\right)^2} e^{-\frac{\left(\overline{s_k^\mu} - s_k^\mu\right)^2}{2\Delta_k^\mu}} ds_k^\mu \forall \omega \in \Omega, \tag{9a}$$

$$\overline{s_k^{\mu}} = \sum_{j \neq i,k}^{N} \overline{w_{ij}} x_i^{\mu*} + u_i^{\mu}, \tag{9b}$$

$$\Delta_k^{\mu} = \sum_{j \neq i,k}^{N} \left( \overline{w_{ij}}^2 - \overline{w_{ij}^2} \right) \left( x_i^{\mu*} \right)^2, \tag{9c}$$

$$\overline{w_{ij}} = \sum_{\omega} \omega P^{\mu} \left( w_{ij} = \omega \right). \tag{9d}$$

See Equations 4–14 in *Molinelli, Korkut, Wang et al. (2013)* for derivation of *Equation 9* and detailed descriptions of the terms in the equation. Note that, BP parameters (temperature factor, $\beta = 2$; complexity factor; $\lambda = 5$; prior information weight, $\kappa = 5$) are adapted for all computations after a systematical analysis of all three parameters (*Figure 4—figure supplement 5*). When prior information is used to construct the models, the complexity term, $\lambda$ is set to 2.5 for phenotypic nodes in order to approximately match the complexity around the phenotypic nodes and rest of the networks. When the parameter space is sufficiently large, the Gaussian approximation significantly reduces the computational complexity of the problem without loss of overall accuracy. By means of the iterative approach with the mean-field Gaussian approximation, the time-complexity of the problem is strongly reduced and the obstacles imposed by combinatorial complexity are circumvented.

### BP-guided decimation
Distinct networks models are instantiated from BP-generated probability distributions with the BP-guided decimation algorithm (*Figure 1—figure supplement 1*) (*Montanari et al., 2007*). This procedure generates distinct and executable network models. In this study, 4000 distinct network models are generated in each computation.

### Simulations with in silico perturbations
Network models are executed with specific in silico perturbations until all system variables {$x_i$} reach steady state. The perturbations acting on node, $i$, are exerted as real-valued, $u_i^{\mu}$, vectors in model *Equation 1*. The DLSODE integration method (ODEPACK) (*Hindmarsh, 1993*) is used in simulations (default settings with, MF = 10, ATOL = 1e-10, RTOL = 1e-20).

## Experimental methods

### Cell cultures and perturbation experiments
All the perturbation experiments are performed using the RAF-inhibitor resistant SkMel-133 cell line. The identity of the SkMel-133 was checked for mislabeling, contamination, and misidentification using a multiplexed PCR/mass spectrometry–based genetic fingerprinting assay run on the HapMap Sequenom platform. In perturbation experiments, the drugs are applied to cell cultures after SkMel-133 cells are grown to about 40% confluence in complete RPMI-1640 medium (10% heat-inactivated fetal bovine serum, 100 units/ml each of penicillin and streptomycin, and incubated at 37°C in 5% $CO_2$) in 6-well plates. 24 hr after drug administration, the perturbed cells are harvested. In control experiments (i.e., no drug condition), cells are treated with the DMSO drug vehicle for 24 hr. SkMel-133 cells are perturbed with 12 targeted drugs applied as single agents or in paired combinations (see *Supplementary file 1A,C* for drug list, presumed targets, dosing and sources). In total, cells are treated with 89 unique perturbations. In paired combinations, each drug concentration is selected to inhibit the readout for the presumed target or the downstream effectors by 40% (IC40) as determined by Western blot experiments (*Molinelli, Korkut, Wang et al., 2013*) (*Supplementary file 1A*). In single agent perturbations, each drug is applied at two different concentrations, IC40 and 2 × IC40. In validation experiments, (+)JQ1 (Cayman Chemicals, Ann Arbor, MI) and the U.S. Food and Drug Administration (FDA)-approved RAFi PLX4032 (Selleckchem, Houston, TX) are used.

### Reverse phase protein arrays
Proteomic response profiles to perturbations are measured using RPPA (MD Anderson Cancer Center RPPA Core Facility) (*Tibes et al., 2006*). The cells are lysed 24 hr after drug treatment. For sample

preparation, cell pellets are lysed in RPPA buffer (1% Triton X-100, 50 mM HEPES pH7.4, 150 mM NaCl, 1.5 mM MgCl$_2$, 1 mM EGTA, 100 mM NaF, 10 mM Na$_4$P$_2$O$_7$, 1 mM Na$_3$VO$_4$, 10% glycerol, freshly added protease and phosphatase inhibitors). After cells are lysed, total protein concentrations in cell lysates are measured with the Bradford assay and the final protein concentrations are adjusted to 1–1.5 mg/ml. Samples are boiled in SDS sample buffer without bromophenol blue for 5 min at 95°C. Cell lysates are spotted on nitrocellulose-coated slides as described in *Tibes et al. (2006)*. Antibody staining intensities are quantified using the MicroVigene automated RPPA module (VigeneTech, Inc., Carlisle, MA) and the standard RPPA protein concentration normalization procedure (*Neeley et al., 2009*) is followed. Three biological replicates are spotted for each sample (i.e., drug condition) on RPPA slides. Each slide is interrogated with a particular antibody, so for each experimental condition 138 proteomic entities (levels of total protein or phoshoprotein) are profiled on 138 slides (*Supplementary file 1C*). The proteomic readouts are log normalized with respect to the corresponding untreated condition readouts. We have eliminated those readouts with intra- or inter-slide coefficient of variation >0.15 (i.e., low reproducibility) and low degree of staining by antibodies. 100 proteomic entities are chosen for further analysis.

## Detection of proteomic entities that respond to perturbations

To generate a constraint set for network inference, we excluded the proteomic entities that do not respond to at least a single perturbation condition from the network models. An iterative signal-to-noise detection algorithm is used for this purpose. At each iteration, the standard deviation of the overall data is computed and data points that deviate for >2.5$\sigma_i$ ($\sigma_i$: standard deviation of all data points at iteration i) from the data average are selected as signal (i.e., responding to perturbation). In the next iteration, the $\sigma_{i+1}$ is computed for all data points except the data points selected as signal in the previous iteration. The $\sigma$ calculation is iterated until convergence. The proteomic entities that respond to at least a single perturbation condition are selected. This has lead to highly reliable response measurements on a final set of 82 proteomic entities, which have been used in the network models.

## Quantitative phenotypic assays

All phenotypic measurements are made in perturbation conditions identical to those of proteomic measurements. Cell viability and cell cycle progression are measured using the resazurin assay (72 hr after drug treatment) and flow cytometry analysis (24 hr after drug treatment), respectively. The percentage of cells in the G1, G2/M, and S phases and sub-G1 fraction are recorded based on the respective distribution of DNA content in each phase. For the resazurin assay, each well is treated with the resazurin (Sigma–Aldrich, St. Louis, MO, Catalog #R7017) at a final concentration of 44 µM for 1 hr. The fluorescent signals are measured at 530-nm excitation wavelength and 590-nm emission wavelength. Standard curves of cell numbers are used to calculate the cell numbers in each sample. The cell cycle progression phenotypes are measured with flow cytometry analysis. Cells are seeded in 6-well plates. 72 hr after drug application, cells are harvested by trypsinization, including both suspended and adherent fractions, and washed in cold PBS. Cell nuclei are prepared as described in *Nusse and Kramer (1984)*, and cell cycle distribution was determined by flow cytometric analysis of DNA content using red fluorescence of 488-nm excited ethidium bromide-stained nuclei. The percentage of cells in the G1, G2/M, and S phases and sub-G1 fraction are recorded.

## Advances in the perturbation biology method

The improvements in perturbation biology have enabled us to establish our method as a widely applicable tool in genomically informed preclinical studies. Here, we obtained signaling models with increased scope (increased coverage of diverse pathways using a data set 20-fold bigger than previously used data sets) and nominated a particular drug combination to overcome the drug resistance in melanoma. The increased scope of the models is particularly critical for biologists to adopt our method as such models provide a nonreductionist insight into relations between the molecular changes and complex biological phenomena such as drug response, cell fate in diverse conditions, or any other cellular phenotype that can be measured quantitatively. Here, we improved the perturbation biology method and solved three key challenges in network inference to reach scopes and predictive power necessary for addressing complex problems in cancer biology.

## A biologically grounded modeling framework that can be incrementally improved

Our current modeling scheme can incorporate both quantitative and qualitative prior information through the introduction of a generalized probabilistic error model to the inference cost function (*Equations 2–7*). In the generalized form of the error model, the prize introduced by the priors is sampled from a probability model with a Gaussian distribution (*Equation 3*) for the strength ($w_{ij}^{prior}$) of each prior interaction. A Boltzmann conversion of the prior probability model leads to an error model that is fully compatible with the BP-generated probability distributions, $P(w_{ij})$. Such a generalized error model will be most useful when quantitative prior information is available for the signaling interactions.

The general error model provides an advantage in building cell-type specific models in diverse but related biological samples. When we build network models for a series of biological systems (e.g., cell lines or patient derived samples) with similar but distinct genomic properties, the BP-generated probability distributions in one cellular system can be extrapolated to another system as a set of probabilistic biases (i.e., priors). The probabilistic extrapolation between different biological samples will significantly reduce the amount of experimental constraints (i.e., perturbation conditions) required to build predictive models while preserving the cell-type specificity for each biological sample studied. This is extremely promising in the context of cancer biology where a model portfolio built from a cohort of patient-derived samples (cell lines or mouse avatars) coupled with genomic profiling can inform treatment decisions for new patients. This bridge between genomic profiles, models, and response to treatment is an open challenge that we think is going to be critical for future precision medicine applications.

The versatile prior error model introduced here enables us to use known biology in various forms that can be available from previous studies or databases. On one extreme, the error model is simplified and reduced to a step function (*Equation 7*). The simplified form of the error model is compatible with the binary priors as are the priors used in this study, which were extracted from the curated signaling databases. Alternatively, one can generate hybrid forms of the cumulative prior error model. Such hybrid models may include (i) the truncated Gaussian model for a subset of priors with quantitative parameters, (ii) the simplified step function form for binary priors, and (iii) alternative forms (e.g., based on a bimodal distribution) for other priors. Although we used the simplified form of the error model and constructed models in a single cell line, we believe the flexibility of the prior error model will be transformative for generating highly predictive models of different biological systems. Such models will subsequently capture the inter-sample variations in response to combinatorial perturbations.

## Systematic integration of prior information

Development of the PERA algorithm leads to systematic extraction and use of prior interactions, which has a series of advantages over ad hoc introduction of priors in the network modeling scheme. First, currently available network inference algorithms and prior network extraction utilities work at either gene or protein level. Therefore, they are of limited use for inference algorithms that operate on phospho-protein levels. PERA allows users to specify both phospho- and total-protein levels of the genes and provide users with prior knowledge with higher granularity. Second, pathway resources are regularly being updated with newly curated interactions; therefore, our prior knowledge on entities of interest is constantly evolving. PERA can work against new versions of pathway data from Pathway Commons and this enables users to obtain up-to-date prior information for their proteins by simply re-running the application. Third, PERA extracts prior information from pathway data integrated from multiple resources in an automated manner; hence, the interactions extracted from the databases are unbiased compared to the prior knowledge obtained from literature by a single researcher. The algorithm also allows users to specify options (such as the level of phosphorylation site mismatch tolerance and the maximum length between two entities to be considered) for adjusting the complexity and the specificity of the final prior network. Here, an open challenge for future is assigning quantitative confidence scores and statistical information for each prior interaction extracted from databases. Currently, the database-derived priors are in binary form. However, with incorporation of literature co-citation data and experimental co-expression data in different tissues will enable us to estimate the confidence scores and statistical parameters for potential prior interactions.

## Performance improvements that can scale the method to our experimental capabilities

We optimized the performance of our algorithm by introducing heuristics at key bottlenecks. This was necessitated by 20-fold increase in data points compared to our previous work. One of the key bottlenecks in the modeling scheme is the BP-guided decimation, for which the BP-derived probability distributions are updated after each step (i.e., after sampling all the BP-derived probability distributions and fixing the interaction value for one edge with a probabilistic draw) (see supplementary *Figure 1—figure supplement 1*). In other words, the posterior probability of a $W_{ij}$ is converted to 1. Not surprisingly, the BP-probability for significant fraction of the selected edges is within the range of 0.95–0.99. As fixing an individual edge from $P^{unfixed}(W_{ij}) = 0.95$ ($P^{unfixed}$: probability before fixing the edge value) to $P^{fixed}(W_{ij}) = 1$ does not significantly alter the probabilistic landscape of the global solution space that involves thousands of other probabilistic events, we updated the global interaction landscape after each decimation step if the $P^{unfixed}(W_{ij}) < 0.95$ for the selected edge. However, accumulation of such residual shifts in individual probabilities may lead to a significant change in the global landscape after consecutive iterations of the decimation algorithm. To prevent such inaccuracy in modeling, we updated the probability landscape after five consecutive decimation steps even if $P^{unfixed}(W_{ij}) > 0.95$ at each step. In practice, the conditional updating scheme described here provided a >threefold increase in the performance of the BP-decimation algorithm without substantially modifying the sampled probability space.

## Acknowledgements

We thank NP Gauthier, P Kaushik, ML Miller, DS Marks, A Braunstein, A Pagnani, R Zecchina, M Cokol, and N Rosen for helpful discussions. We thank G Mills for critical reading of the manuscript, helpful discussions, and help with the acquisition of the RPPA data. This work was funded in part by the Center for Cancer Systems Biology grant U54 CA148967 (NIH), the National Resource for Network Biology grant GM103504 (NIH), the Research Resource for Biological Pathways grant U41 HG006623 (NHGRI), and a Melanoma Research Alliance Established Investigator Award.

## Additional information

### Funding

| Funder | Grant reference | Author |
|---|---|---|
| National Institutes of Health (NIH) | The National Resource for Network Biology grant, GM 103504 | Anil Korkut, Emek Demir, Özgün Babur, Chris Sander |
| National Human Genome Research Institute (NHGRI) | The Research Resource for Biological Pathways grant, U41 HG006623 | Emek Demir, Özgün Babur, Chris Sander |
| Melanoma Research Alliance (MRA) | Established Investigator Award | Anil Korkut, Weiqing Wang, Chris Sander |
| National Institutes of Health (NIH) | Center for Cancer Systems Biology grant, U54 CA148967 | Anil Korkut, Weiqing Wang, Chris Sander |

The funders had no role in study design, data collection and interpretation, or the decision to submit the work for publication.

### Author contributions

AK, Developed the network inference algorithm, conceived and designed the experiments, developed the PERA algorithm, analyzed the data, developed and optimized the code, wrote the paper; WW, Conceived and designed the experiments, performed the experiments, analyzed the data; ED, Developed the PERA algorithm, analyzed the data, wrote the paper; BAA, Developed the PERA algorithm, developed and optimized the code; XJ, Performed the experiments; EJM, Developed the network inference algorithm, developed and optimized the code; ÖB, Developed the PERA algorithm; DLB, Analyzed the data, coordinated the research activities, revised the paper; SOS, Analyzed data, designed the website for the perturbation biology method and for

interpretation of the data and results in this study; DBS, Contributed reagents/materials/analysis tools; CAP, Conceived and designed the experiments, contributed reagents/materials/analysis tools, analyzed and discussed the results; CS, Conceived and designed the experiments, developed the PERA algorithm, developed the network inference algorithm, analyzed the data, wrote the paper

## Additional files

### Supplementary file

• Supplementary file 1. (**A**) Drugs used in perturbation experiments. (**B**) Proteins that respond to at least one perturbation and exist in models. (**C**) Perturbation conditions. (**D**) Predicted G1-arrest response to combined targeting of c-Myc and other nodes (Perturbations on c-Myc + X). (**E**) Statistical validation of perturbation models. (**F**) Predicted co-targeting strategies in SkMel133 cell line.

### Major dataset

The following dataset was generated:

| Author(s) | Year | Dataset title | Dataset ID and/or URL | Database, license, and accessibility information |
|---|---|---|---|---|
| Wang W, Korkut A, Sander C | 2015 | Phosphoproteomic response to targeted drug combianations in SkMel133 cells | http://www.sanderlab.org/pertbio/ | Publicly available online. |

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
