## [Decision Letter]

Thank you for sending your work entitled “Perturbation biology models predict c-Myc as an effective co-target in RAF inhibitor resistant melanoma cells” for consideration at *eLife*. Your article has been favorably evaluated by Aviv Regev (Senior editor), a Reviewing Editor, and two additional reviewers.

The Reviewing editor and the reviewers discussed their comments before we reached this decision, and the Reviewing editor has assembled the following comments to help you prepare a revised submission.

The editorial team has identified two areas of major concern: (1) novelty of the approach and (2) biological significance.

Regarding novelty of the approach, a portion of the most interesting methodological innovations has been already published by a subset of the authors. The pre-processing PERA step is new, and so are a number of other technical advances. In this paper, the authors extended the scope of the analysis substantially (phosphoprotein antibodies from 16 to 138, perturbation conditions from 44 to 89, and phenotypes from 1 to 5), and we appreciate that this extension is critical for biologists to adopt such approaches. However, the authors fail to clearly present the innovations over previous work. The authors should make an effort, perhaps in a dedicated subsection under Methods, to offer a nuanced discussion of the approach stating how it supports scientific discovery, what is novel relative to the earlier studies, and what are the open challenges. Overall the approach is technically solid and well motivated, and *eLife* readers will benefit from such a discussion in the context of the new and extensive data set.

Regarding biological significance, the reviewers all agree that the authors do not clearly address how their results advance our understanding of the biology, nor provide any novel biological insights. It is very important that any conceptual novelty is highlighted in the revised manuscript.

Other serious concerns include limitations of the approach with respect to predictions (reviewer 2, please address this in the response and in the manuscript), and lack of a convincing statistical validation (e.g., frequentist coverage of the predictions in a realistic simulation study – where data are generated using parameters estimated from the real data). We would like to see these addressed in the revision as well.

Overall, this is a solid piece of work, and we look forward to receiving a revision.

Reviewer #1:

The authors applied belief propagation to data from systematic pharmacological perturbations and generated network models of signaling in melanoma cells. The authors used these models to predict cellular responses to untested drug perturbations. Simulating the models, the authors made predictions for effective combinations of perturbation. The authors argue that one of these predictions, that co-targeting c-Myc with MEK or RAF is synergistic, is non trivial and go on to demonstrate experimentally a synergistic effect of co-treating.

Conceptual and high-level feedback:

In my opinion, the most significant and general result of this work is showing the promise of genomically-informed preclinical trials and providing a concrete example of a synergistic pharmacological treatment inhibiting the cell division of melanoma cells. The experimental design and the network inference methodology seem careful and well executed. The results appear convincing. On the down-side, I do not see much conceptual novelty or results contributing to our understanding of the biology. It is also not clear to me that the initial synergistic effect of co-targeting, which is interesting and explored in details, will actually prevent the development of drug-resistance over the long term in a clinical application. I invite the authors to emphasize any conceptual advances in their work and new biological conclusions that I may be missing. I think that conceptual advances would contribute the most to elevating the significance and general interest in this work.

[Minor comments not shown.]

Reviewer #2:

In this study, the authors use a belief propagation (BP) algorithm to construct a network model of nearly 100 (phospho)proteins and several phenotypes in response to single and paired drug treatments. The network models use a differential equation framework, so they can be simulated to predict the response of the observed variables (proteins, phenotypes, etc.) to new perturbations and arbitrary combinations of perturbations. Simulations nominate c-Myc, which was not perturbed in the original drug screen, due to its in silico effect on G1 arrest. c-Myc is targeted indirectly with JQ1, and JQ1 in combination with predicted complementary inhibitors (MEKi and RAFi) induce strong changes in viability and G1 arrest.

The modeling approach is inherently limited to make predictions about proteins on the protein array in the perturbation screen and edges represent indirect effects, but the experimental/computational technique – in particular the focus on quantitative, predictive instead of descriptive networks – is powerful when there is sufficient coverage on the protein array. The algorithmic innovations have already been published (BP algorithm and preliminary SkMel-133 analysis in Molinelli 2013; including prior knowledge in Miller 2013). This manuscript expands the (phospho)protein antibodies (16 to 138), perturbation conditions (44 to 89), and phenotypes (1 to 5) modeled in RAFi-resistant SkMel-133 cells and evaluates new predicted pairs of inhibitors. It also introduces the pre-processing PERA step, which improves the inclusion of prior pathway knowledge by making it more systematic. The prior implementation – using soft constraints and rewarding edges with prior support instead of penalizing those without – is well-motivated given that interactions in pathway databases are derived from different cell types.

1) The criteria used to select c-Myc and G1 arrest instead of other phenotypes and top-ranked in silico perturbations are not made explicit, making it difficult to assess how well the method could predict additional novel perturbations. Figure 5—figure supplement 2 shows that the top perturbation for all phenotypes is not novel but rather one of the activity nodes for the originally screened drugs. c-Myc does cause a substantial change to G1 arrest in silico, but so do other novel perturbations like p38 on viability, STAT3pY705 on S arrest, etc. Furthermore, it appears that there was no preference given to combinations of drugs that are predicted to be synergistic (similar to Miller 2013) or take effect at the lowest possible doses. Indeed, the top viability perturbations – PKCi and CDK4i – only reduced viability at high doses.

2) Is the RAFi-resistance phenotype of the SkMel-133 cell line recapitulated in the computational model? bRAF or aBRAF are top 10 predicted perturbations for all phenotypes except G2 arrest.

3) Are the observed changes in G1 arrest under inhibitor treatment statistically significant?

4) What was done to verify the identity of the SkMel-133 cell line?

5) Edge frequencies are one way to assess BP stability, and Figure 4—figure supplement 2 shows that there are few stable edges. How much do the phenotype predictions vary across the 4000 simulations?

[Minor comments not shown.]

---

## [Author Response]

*The editorial team has identified two areas of major concern: (1) novelty of the approach and (2) biological significance*.

We included two new sections to address these concerns (please see below).

*Regarding novelty of the approach, a portion of the most interesting methodological innovations has been already published by a subset of the authors. The pre-processing PERA step is new, and so are a number of other technical advances. In this paper, the authors extended the scope of the analysis substantially (phosphoprotein antibodies from 16 to 138, perturbation conditions from 44 to 89, and phenotypes from 1 to 5), and we appreciate that this extension is critical for biologists to adopt such approaches. However, the authors fail to clearly present the innovations over previous work. The authors should make an effort, perhaps in a dedicated subsection under Methods, to offer a nuanced discussion of the approach stating how it supports scientific discovery, what is novel relative to the earlier studies, and what are the open challenges. Overall the approach is technically solid and well-motivated, and* eLife *readers will benefit from such a discussion in the context of the new and extensive data set*.

We included a detailed discussion of the methodological innovations, open challenges and significance of the method for scientific discovery. This is a separate section titled “Advances in the perturbation biology method” in Methods.

*Regarding biological significance, the reviewers all agree that the authors do not clearly address how their results advance our understanding of the biology, nor provide any novel biological insights. It is very important for* eLife *that any conceptual novelty is highlighted in the revised manuscript*.

We rewrote a significant portion of the Discussion to better address the significance of our biological findings and their importance for cancer research, melanoma and drug resistance. In this section, we summarized:

A) The previous attempts (drug combinations) to overcome the drug resistance in melanoma.

B) Why targeting c-Myc and the ERK pathway is potentially more beneficial than other combinatorial interventions in melanoma.

C) A more generalized view of overcoming drug resistance by co-targeting upstream genomic aberrations and downstream signaling proteins, which are connected to diverse pathways.

D) The details of our findings in terms of synergistic response to BET bromodomain inhibitors and ERK pathway inhibitors.

E) Potential clinical applications of BET bromodomain inhibitors either as a single agent or in combination with kinase inhibitors (e.g., RAF-inhibitor) in cancer therapy.

*Other serious concerns include limitations of the approach with respect to predictions (reviewer 2, please address this in the response and in the manuscript), and lack of a convincing statistical validation (e.g., frequentist coverage of the predictions in a realistic simulation study – where data are generated using parameters estimated from the real data). We would like to see these addressed in the revision as well*.

For limitations of the approach with respect to predictions, please see the response to reviewer 2.

For statistical validation, we computed the frequentist coverage probabilities of our predictions (Figure 3, Figure 3—figure supplement 1 and [Supplementary-material SD1-data]) for all cross validation predictions (CVs). First, we expanded the leave-11-out cross-validation (CV) simulations from 2 (AKTi and RAFi in Figure 3) to all possible conditions (2+10 calculations, Figure 3—figure supplement 1). Next, for each cross-validation/simulation set, we provided the frequentist coverage probabilities in 95%, 90% and 70% confidence intervals (CI) ([Supplementary-material SD1-data]). The statistical validations show that in 10 out 12 of leave-11-out CVs, the frequentist coverage is > 95%. For CV that involves the prediction of response to HDACi, the frequentist coverage probability is 93% for the 95% CI. We concluded that this is also a high-quality prediction given the level of ambition in our cross validation scheme. However, the frequentist coverage for CDK4i predictions is below 95% (88%). Indeed, the correlation coefficient between the simulated values and the hidden experimental conditions was also ∼0.70 for CDK4i, while the correlation coefficients are as high as 0.88 for other inhibitors. This is most probably due to the fact that the CDK4 inhibitor was highly toxic compared to the other compounds. The number of viable cells under CDK4i was lower than those in other perturbation conditions. Therefore, the prediction quality for CDK4i cross-validation was lower compared to the other CVs. Although we state that a predictive power of 0.70 is still sufficient to make useful predictions, this demonstrates the importance of a high quality training dataset for reaching excellent predictive power such as in all other conditions.

Overall, this is a solid piece of work, and we look forward to receiving a revision.

Reviewer #1:

*Conceptual and high-level feedback*:

*In my opinion, the most significant and general result of this work is showing the promise of genomically-informed preclinical trials and providing a concrete example of a synergistic pharmacological treatment inhibiting the cell division of melanoma cells. The experimental design and the network inference methodology seem careful and well executed. The results appear convincing. On the down-side, I do not see much conceptual novelty or results contributing to our understanding of the biology. It is also not clear to me that the initial synergistic effect of co-targeting, which is interesting and explored in details, will actually prevent the development of drug-resistance over the long term in a clinical application. I invite the authors to emphasize any conceptual advances in their work and new biological conclusions that I may be missing. I think that conceptual advances would contribute the most to elevating the significance and general interest in this work*.

We thank the reviewer for this suggestion and we believe this will certainly increase the significance of our study. As stated above, we have included two separate sections to address this issue: (1) we updated the Discussion section and described in detail the biological significance of our findings and the importance of the discovered combination strategy for RAFi resistance problem in melanoma biology; (2) we included another section to the Methods, where we discuss in detail the methodological improvements in this paper, how it can accelerate the biological discoveries and what the current open challenges are.

Reviewer #2:

*1) The criteria used to select c-Myc and G1 arrest instead of other phenotypes and top-ranked in silico perturbations are not made explicit, making it difficult to assess how well the method could predict additional novel perturbations.*
Figure 5—figure supplement 2
*shows that the top perturbation for all phenotypes is not novel but rather one of the activity nodes for the originally screened drugs. c-Myc does cause a substantial change to G1 arrest in silico, but so do other novel perturbations like p38 on viability, STAT3pY705 on S arrest, etc. Furthermore, it appears that there was no preference given to combinations of drugs that are predicted to be synergistic (similar to Miller 2013) or take effect at the lowest possible doses. Indeed, the top viability perturbations – PKCi and CDK4i – only reduced viability at high doses*.

We ranked the perturbation conditions based on their predicted influence on each phenotype in the model. We selected the top ranked perturbations as candidates for further experimental validation. One of the strengths of our method is that there is not a single but a set of useful (top ranked) predicted combinatorial perturbations in a given context (e.g., RAFi resistant melanoma cells). Each of these predictions can be tested using targeted drugs or other systems such as shRNA, CRISPR-Cas. We believe that these predictions will be a very useful resource for experimental biology groups that aim to test drug combinations in melanoma. We now have provided the predicted response to all the combinatorial perturbations on our webpage. Biologists can explore the predicted responses to > 70000 combinatorial perturbations (combinations of different doses, strengths etc.) for five phenotypes. We also provide all of the model solutions so that they can predict response to alternative combination schemes (e.g., high order combinations, time staggered combinations etc.) using any type of program that can execute ODEs.

In this study, we chose to experimentally test one of the most promising predicted combinations: c-Myc/ERK-pathway (MEKi or RAFi) and G1 arrest. Reasons for selecting this combination include: (1) we predicted a substantial synergy for these combinations and these are the top responding targets for G1-arrest; (2) the combinations involve targets (e.g., ERK pathway) that are known targets for treatment of melanoma (3) induction of G1-arrest is a promising strategy for controlling melanoma growth (Solit et al., Nature, 439, 358-362); and various genes and cell cycle pathways associated with G1-arrest are involved in melanoma progression and RAF-inhibitor resistance (51; 67). Such genes involve RB, CDKN2A & CDKN2B (deleted in SkMel133 cells), CDK4 and many others (4) the predicted targets are potentially druggable, albeit indirectly in the case of c-Myc. We provide further details related to other predicted combinations below.

There are certainly other very promising predictions, such as targeting p-cJUN (Using JNK inhibitors) and ERK pathway for G1-arrest.

Some of the predicted perturbations/targets (activity nodes) were already included in the experimental training set. Combinatorial perturbations that involve such trivial and other novel/nontrivial targets should still be tested in the laboratory. One example is the HDAC inhibitor that induces a strong G2-arrest. It will be interesting to search for useful combinations involving HDACi. Some predicted targets are not “activity nodes” (e.g., STAT3pY705) and are less interesting than others since they are highly related to the activity nodes (i.e., perturbations). For example, STAT3i is already in the experimental set and it reduces both STAT3 phosphorylation (STAT3pY705) and cell viability. Therefore it was not an immediate choice for us. Interestingly p53 and aMDM2 have very similar G2-arrest responses (Figure 5—figure supplement 2). Note that MDM2 is a p53-specific ubiquitin ligase and controls p53 stability.

Some of the predicted targets are not directly druggable with targeted compounds (e.g., most total level targets, such as p38/MAPK14, p27/Kip1, CRAF, TSC2, caveolin). New gene editing technologies such as CRISPR may be useful to experimentally perturb these nodes and test our predictions.

We provide a very brief summary of the above in the revised manuscript (Results, last paragraph). We also included a table that lists the top predictions ([Supplementary-material SD1-data]).

2) Is the RAFi-resistance phenotype of the SkMel-133 cell line recapitulated in the computational model? bRAF or aBRAF are top 10 predicted perturbations for all phenotypes except G2 arrest.

The RAF-inhibitor resistance was recapitulated in the computational models. A good measure of the RAF-inhibitor resistance in the models is the response to perturbation on the aBRAF but not the BRAF. The BRAF node corresponds to the total protein level of the BRAF protein and aBRAF is an activity node that couples the effect of the RAF inhibitor to rest of the network.

aBRAF is in the top 10 predictions only for G1-arrest. The experimental phenotypic response data (Figure 2) show a slight G1-arrest response to MEK and RAF inhibitors (Note that RAFi/MEKi combination does not lead to a drastic increase G1-arrest. Also see the strong G2-arrest response to HDACi, which was recapitulated in the models; Figure 5—figure supplement 2). These effects are recapitulated in the models as aBRAF is in top 10 for the G1-arrest prediction. However, as can be seen in Figure 6, the response to combined targeting of c-Myc, CyclinD1, aMEK, aBRAF and p-cJun is higher than the single agent perturbations. aBRAF was not in the top10 predictions for other phenotypes.

The total BRAF level is in the top 10 for cell viability, S-arrest and G2M nodes but the predictions do not suggest that BRAF has a drastic effect on any of the phenotypes (compared to the top few predictions). According to the vast literature on RAF inhibitors and melanoma, the total BRAF level usually does not immediately change in response to RAF inhibition and is not a direct measure of RAFi resistance, as RAFi inhibits phosphorylation of MEK by mutated BRAF protein. However, it is very interesting that BRAF (and CRAF) levels are related to cell viability and some of the cell cycle phenotypes. It is reported that the long-term changes in expression levels and heterodimerization of BRAF and CRAF have nontrivial implications in the emergence of resistance to RAF inhibitors (See Figure 2 in Lito, Rosen and Solit, 2013, Tumor adaptation and resistance to RAF inhibitors, Nature Medicine). Due to the transactivation of the wild-type BRAF and CRAF and RAF isoform switching in response to RAF inhibition, the relation between the (B/C)RAF concentration and drug resistance is not linear. Experimentally testing the effect of changes in BRAF/CRAF on RAF inhibitor-induced phenotypes and RAFi sensitivity will be very interesting. On the other hand, we did not perform experimental tests related to predictions on RAF levels. In the current manuscript, it would not be more than a speculation to relate (B/C)RAF levels and drug resistance.

3) Are the observed changes in G1 arrest under inhibitor treatment statistically significant?

Cell cycle arrest experiments (flow cytometry) were run in triplicates. Although we refrained from running a t-test with three data points, we provided standard error bars in Figure 6. The error bars indicate that the G1 arrest response for the combination is significantly stronger than those for other conditions (no drug and single drug perturbations).

4) What was done to verify the identity of the SkMel-133 cell line?

Our collaborators and co-authors, David Solit and Christine Pratilas, verified the identity of the SkMel133 and a panel of other melanoma cell lines. They checked for mislabeling, contamination and misidentification using a multiplexed PCR/mass spectrometry (MS)–based genetic fingerprinting assay run on the HapMap Sequenom platform. The method is described in: Janakiraman et al., Genomic and Biological Characterization of Exon 4 KRAS Mutations in Human Cancer, Cancer Res. 2010 Jul 15;70(14):5901-11. We now state this in the Methods section of the revised manuscript.

*5) Edge frequencies are one way to assess BP stability, and*
Figure 4—figure supplement 2
*shows that there are few stable edges. How much do the phenotype predictions vary across the 4000 simulations?*

We now provide the distributions of all the predicted responses over 4000 solutions on our webpage (http://130.211.159.201/pertbio/). The statistical information for each distribution is also provided on the webpage.